# Integrative spatial omics reveals distinct tumor-promoting multicellular niches and immunosuppressive mechanisms in Black American and White American patients with TNBC

Racial disparities in the clinical outcomes of triple-negative breast cancer (TNBC) have been well-documented, but the underlying biological mechanisms remain poorly understood. To investigate these disparities, we employed a multi-omic approach integrating imaging mass cytometry and spatial transcriptomics to characterize the tumor microenvironment (TME) in self-identified Black American (BA) and White American (WA) TNBC patients. Our analysis revealed that the TME in BA patients is marked by a network of endothelial cells, macrophages, and mesenchymal-like cells, which correlates with reduced patient survival. In contrast, the WA TNBC microenvironment is enriched in T-cells and neutrophils, indicative of T-cell exhaustion and suppressed immune responses. Ligand-receptor and pathway analyses further demonstrated that BA TNBC tumors exhibit a relatively "immune-cold" profile, while WA TNBC tumors display features of an "inflamed" TME, suggesting the evolution of a unique immunosuppressive mechanism. These findings provide insight into racially distinct tumor-promoting and immunosuppressive microenvironments, which may contribute to the observed differences in clinical outcomes among BA and WA TNBC patients.

A profound racial disparity has been identified between Black American (BA) women and their White American (WA) counterparts with respect to the incidence[1–3] and clinical trajectory[2,3] of breast cancer (BCa). Specifically, BA women develop BCa at a relatively younger age[4], have a two-fold higher chance of developing Triple Negative (TN) tumors[5], with more aggressive disease features (early onset, higher tumor grade, larger tumor size, ER-negative status, distant metastasis)[6]. In addition, the incidence and mortality of the particularly aggressive triple-negative breast cancer (TNBC) subtype are higher in Black women than in White women[7,8]. Importantly, cancer-related disparities in mortality between BA and WA women was higher for TNBC than other subtypes, even after adjusting for age, stage, treatment, socio-economic status, poverty index, and treatment delay[9,10]. Genetic, environmental, and healthcare access/utilization factors may all contribute to this disparity[11]. Many studies have examined the genetic component to understand the biological underpinnings of racial disparity[12].

Studies examining global discovery of gene expression signatures that distinguish BA vs WA TNBC tumors have reported distinct tumor-associated immunologic profiles in BA patients[13–15] and in patients of African descent[16,17]. The bulk-level RNA sequencing approach used in these studies revealed heterogeneity in TNBC and provided cues on

✉ e-mail: qian.zhu@bcm.edu; arun.sreekumar@bcm.edu

the involvement of unique cellular-level interactions in the tumor microenvironment (TME)[18], which were not defined. Notably, variations in the composition of the TME between BA and WA BCa patients have been reported[19,20], wherein computational deconvolution of bulk samples has predicted the existence of cell types in various combinations in spatial niches. These include Tregs and naïve B cells that have been shown to exhibit positive correlations with African ancestry[16], while activated mast cells negatively correlated with African ancestry[16]. Further, the level of tumor-associated lymphocytes was found to be similar between BA and WA TNBCs[13]. Controversies regarding the composition of TME and cellular interactions in BA and WA TNBCs could not be resolved due to the limitations of bulk sequencing and the reliance on computational deconvolution. We believe that predicted cell types could exist in multiple different combinations, in distinct spatial niches, that could be associated with differences in clinical outcomes in TNBC. Hence, dissecting the spatial granularity in niches within the TME would be of paramount importance to obtain deeper insights into the altered biology associated with BA and WA TNBC.

While recent studies employing spatial proteomic[21,22] and transcriptomic[23] technologies have provided insights into the molecular, cellular, and spatial phenotypes governing cancer metastasis[24] and tumor recurrence[25], they have not been extended to the study of BA and WA TNBCs. Previous multi-omics experiments have facilitated the creation of predictors for treatment response and enhanced our understanding of the biological mechanisms behind molecular cancer phenotypes[26–28]. Basic spatial profiling of proteins demonstrated differences between BA and WA BCa patients, despite that these differences were not associated with survival[29]. Moreover, spatial transcriptomics analysis demonstrates that there are racial differences in hypoxic tumor content and regions of immune-rich infiltrates among those with TNBC[30]. However, the composition and spatial heterogeneity of these immune-rich infiltrates remain unclear. Furthermore, whether racial differences exist in the spatial interactions between various components within the TME and their association with disease outcomes remains unknown. Although imaging mass cytometry (IMC)[21], multiplexed ion beam imaging (MIBI)[22], and single-cell RNA sequencing[31] have been used to characterize the TME of TNBC, all patients included in these studies were of European descent.

The multifactorial social construct, race, encapsulates both genetic ancestry and the lived environment. Further, we posit that the experience of racism and being a member of a racialized group differentiates the lived experience of BA and WA women in multiple ways, all of which together result in biological differences. In this study, we examine these biological differences that may be especially pertinent to understanding the poorer outcomes among BA compared to WA. To date, much of the research examining these biological factors has been done on WA women. An unintended consequence of this is that it can lead to improving treatment options and therapy that are largely developed to address the health of WA. This results in widening disparities in outcomes as we become better at treating BCa in WA, but not BA[32]. Indeed, as BCa treatment and therapy options have increased over the last three decades, resulting in better BCa outcomes overall, there has also been a stark increase in the inequity of BCa outcomes between WA and BA women over this same time period[33]. Understanding the differences in the molecular and cellular interactions in BA women is pertinent to ensuring that racial disparities in outcomes do not widen further. We believe that a spatial multi-omics study of a racially diverse TNBC cohort, as described here, should permit one to understand the biological factors underpinning the large survival gap currently experienced by BA patients with TNBC, while also laying the foundation to develop therapeutic interventions that are inclusive of BA women with BCa. In this spatial multi-omics study, we uniquely conduct an integrative analysis combining information obtained from spatial single-cell level IMC[34] and re-analysis of spatial transcriptomics data[30] to uncover racially enriched spatial cell-cell interactions and the larger cell communities that define unique BA and WA TNBC-associated niche. These cell-cell interactions and race-specific niche correlate with patient survival and infer molecular characteristics of WA and BA tumors with a spatial resolution. IMC data measured the expression of 30 proteins at the single cell level, which was then integrated with existing 10X Visium ST data[30], followed by validation of the findings using an independent IMC, 10X ST and Nanostring region of interest (ROI)-based transcriptomics analysis.

Our results uniquely highlight the existence of multicellular niches and immunosuppressive mechanisms involving tumor cells, immune cells, and endothelial cells in TNBC that differ between BA and WA. Of note, these insights were not discernible from previous bulk-level RNA sequencing studies, as well as from unimodal studies of spatial proteomics or transcriptomics, thus underscoring the importance of spatial multi-omics integration to understand race-specific biological mechanisms.

## Results

### Description of the multi-omic cohort, discovery, and validation datasets

We collected triple-negative breast cancer (TNBC) patient specimens from multiple sources and divided them into discovery and validation cohorts (Fig. 1a). Imaging mass cytometry (IMC) was performed on a discovery cohort of 57 patients (26 BA and 31 WA) from Baylor Scott and White Hospital (BSW), followed by validation in an independent cohort of 46 patients (15 BA and 31 WA) from Roswell Park Comprehensive Cancer Center (Roswell Park) and an additional 10 patients (5 each BA and WA) from BSW (BSW2). For spatial transcriptomics (ST), 10X Visium ST was conducted on 9 patients (4 BA and 5 WA) from BSW2, and the data were integrated with Visium ST results from 10 patients (6 BA and 4 WA) from a community hospital in Georgia (PI: Dr. Aneja) and 20 patients (10 each BA and WA) from a previously published TNBC ST dataset by Bassiouni et al. (BAS). The BAS dataset was utilized for initial discovery, while the BSW2 and Georgia datasets served for validation of BA and WA-specific spatial niche gene signatures. To further support our findings, Nanostring GeoMX DSP was applied to the BSW-Discovery cohort for region-of-interest (ROI) and compartment-based mRNA assessments, providing additional validation for BA and WA-associated niche gene signatures. In total, this data collection effort resulted in spatial immune proteomics data from 113 TNBC patients and spatial gene expression data from 39 TNBC patients, enabling robust identification of spatial niches associated with tumor behavior disparities between BA and WA TNBC patients. Figure 1b summarizes the analyses conducted, which include identification of cell-cell interactions and community detection using IMC data, assessment of the prognostic significance of cell communities, spatial localization of cell communities in Visium ST, determination of extended gene signatures (ESG), and ligand-receptor analysis to reveal underlying drivers within the spatial niches.

### Overview of IMC discovery set

We examined a racially balanced and clinically matched cohort of 57 surgically resected TNBC tissue samples, comprising 26 self-reported BA and 31 self-reported WA women (see Supplementary Table 1 for clinical details, and Supplementary Fig. 1 for H&E images). Self-reported race provides a more accurate representation of individuals' lived experiences, which may influence the biological factors contributing to cancer-related health disparities. All patients in the cohort had survival outcome data available for up to 10 years. There was no statistically significant difference in overall survival between BA and WA patients in this cohort (Table 1; univariate analysis for race). In addition, the mutational status of PIK3CA and TP53[8] did not differ significantly between BA and WA patients in our study (Supplementary Table 1).

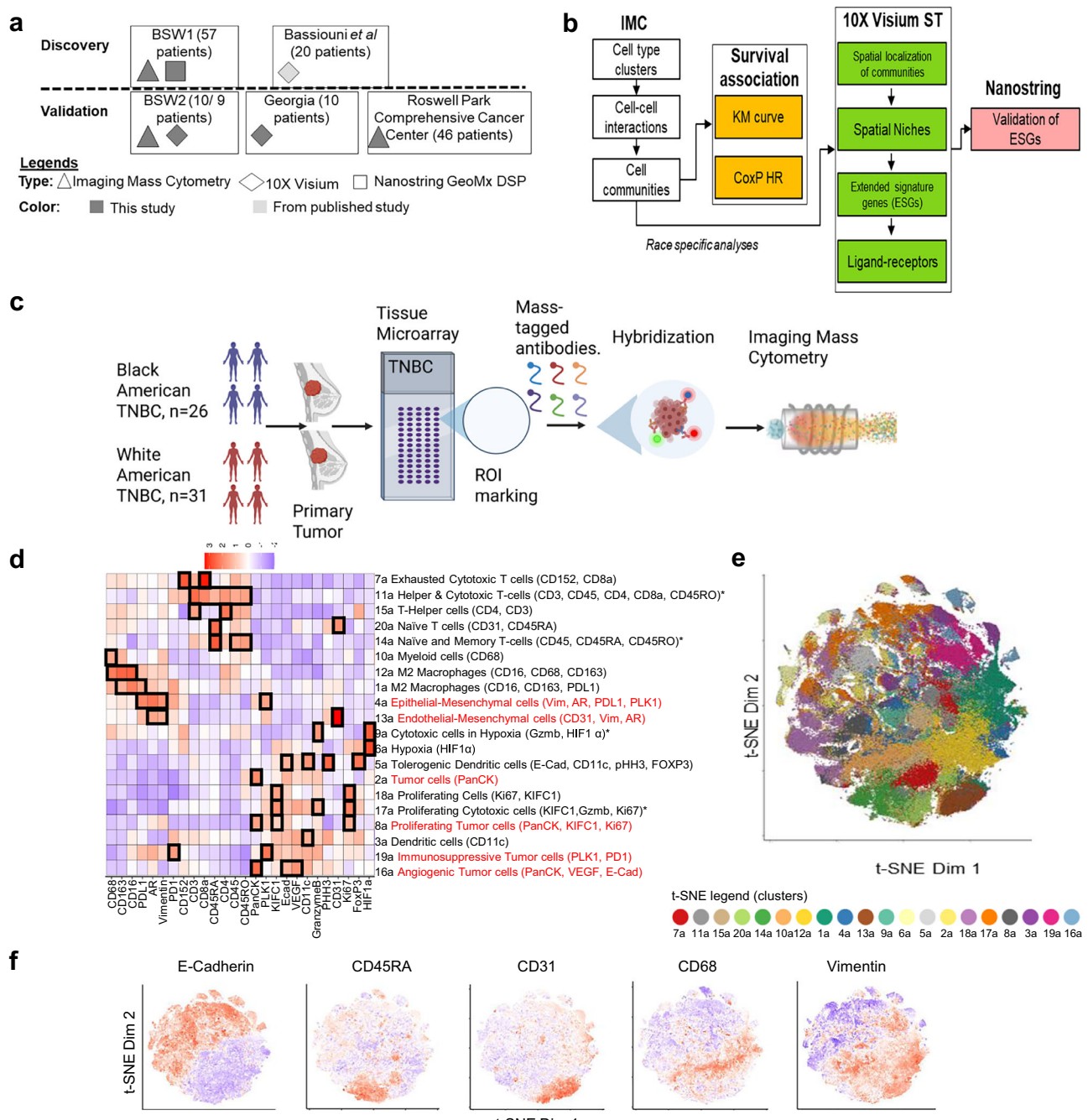

**Fig. 1 | Imaging Mass Cytometry (IMC) Profiling of Self-Reported Black American (BA) and White American (WA) Triple Negative Breast Cancer (TNBC).** **a** Schematic representation of the clinical cohorts analyzed, and the corresponding types of analyses performed for each group. For the Baylor Scott & White (BSW)2 validation cohort sample numbers 10/9 describe samples examined using imaging mass cytometry and 10X spatial transcriptomics, respectively. **b** Overview of the analytical workflow employed throughout the study. **c** Diagram illustrating the imaging mass cytometry-based immune cell profiling process applied to BA and WA TNBC tumors, utilizing a tissue microarray (TMA) format. ROI: Region Of Interest.

Figure created in BioRender. Sreekumar, A. (2025) https://BioRender.com/c0tobe3. **d** Unsupervised clustering of individual cells, segmented from IMC data of BA and WA TNBC tumors, identified 20 distinct clusters. The differential protein markers and their corresponding nomenclature (columns) across these 20 clusters (rows) are shown. Source data are provided in Github (10.5281/zenodo.15353111). **e** Representative t-SNE plots illustrating the distribution of the 20 clusters as described in panel (**d**) within the expression space. **f** Overlay of expression patterns for E-Cadherin, CD45RA, CD31, CD68, and Vimentin onto the t-SNE plots.

To investigate the single-cell spatial interaction landscape, we performed IMC on multiple ROIs (see Supplementary Fig. 2) selected from individual tumors, which were arranged in a tissue microarray (TMA) format (each core with a diameter of 3 mm), using a panel of 26 antibodies targeting immune-regulatory, stromal, and epithelial proteins (overview in Fig. 1c). ROIs were chosen based on H&E-stained images by a breast pathologist and were subsequently confirmed by a

second pathologist. The selected ROIs included areas containing tumor cells intermixed with tumor microenvironment (TME)-related immune and stromal cells. These ROIs were selected from both the tumor center and periphery, categorized as either immune-rich or immune-poor (Supplementary Fig. 3). An equal number and category of ROIs were chosen from tumors of BA and WA patients (Supplementary Table 1).

**Table 1 | Cox Proportional Hazard Analysis for the Baylor Scott and White Discovery Data.** *P*-values: Wald test, 1-sided

| | Hazard Ratio (HR) | P-value |
|---|---|---|
| **Univariate model** | | |
| Race | 1.55 | 0.5 |
| Stage | 1.76 | 0.14 |
| Grade | 0.82 | 0.35 |
| Age | 0.98 | 0.59 |
| Body Mass Index (BMI) | 1.02 | 0.6 |
| Diabetes Mellitus Status | 0.86 | 0.85 |
| PIK3CA Mutation Status | 0.75 | 0.53 |
| RAS Mutation Status | 0.40 | 0.2 |
| TP53 Mutation Status | 0.40 | 0.18 |
| BA-communities-all | 3.45 | 0.06 |
| WA-communities-all | 1.18 | 0.40 |
| **Multivariate model** | | |
| *Adjusting for age:* | | |
| BA-communities-all | 3.38 | 0.065 |
| WA-communities-all | 1.15 | 0.417 |
| *Adjusting for race:* | | |
| BA-communities-all | 3.73 | 0.066 |
| WA-communities-all | 1.41 | 0.315 |
| *Adjusting for stage:* | | |
| BA-communities-all | 4.66 | 0.037 |
| WA-communities-all | 1.02 | 0.49 |
| *Adjusting for age, race, and stage:* | | |
| BA-communities-all | 5.35 | 0.037 |
| WA-communities-all | 1.21 | 0.397 |
| *Adjusting for age, race, BMI, stage:* | | |
| BA-communities-all | 7.69 | 0.021 |
| WA-communities-all | 1.27 | 0.368 |

We initially segmented over 270,000 single cells based on these ROIs. Through unsupervised clustering with multiple initializations, we identified 16 distinct single-cell types and four multi-cell type clusters based on marker protein expression (Fig. 1d, clusters 1a to 20a; multi-cell type clusters indicated by asterisk). Differential analysis revealed unique protein expression profiles within each of the 20 clusters, which were named according to co-expressed markers (e.g., Exhausted Cytotoxic T cell, Cluster 7a, characterized by CD152 and CD8a). Six clusters were associated with tumor characteristics (Fig. 1d, red fonts), while 14 clusters were associated with immune profiles (Fig. 1d). Immune clusters covered diverse cell types such as exhausted T cells (cluster 7a), helper T-cells (15a), naïve T cells (20a), macrophages (12a, 1a), and dendritic cells (5a and 3a). Tumor compartment covered cells of diverse states, such as proliferative (18a), hypoxic (9a), mesenchymal (4a), and angiogenic (16a) states. The t-SNE plot clearly demarcates the clusters (Fig. 1e), and Fig. 1f overlays the expression of E-Cadherin, CD45RA, CD31, CD68, and Vimentin on the t-SNE, highlighting distinct clustering patterns.

### Spatial cell-cell interactions, rather than cluster abundances, distinguish BA and WA TNBC

We analyzed cluster abundances across tumors from patients of both racial groups. While most clusters exhibited similar abundances, several were enriched in specific patient subsets based on race (Supplementary Fig. 4a). For example, naïve T cells, represented by Cluster 20a (CD31, CD45RA), were more abundant in BA TNBC, whereas exhausted cytotoxic T cells, identified by Cluster 7a (CD152, CD8a), and were more prevalent in WA TNBC. However, due to the unique marker

combinations expressed in each patient's tumor, cluster abundance alone was insufficient to segregate patients based on self-reported race (Supplementary Fig. 4b).

Given that differences in relative cluster abundances could not fully explain the racial disparities in BA and WA TNBC, we hypothesized that spatial cell-cell interactions might better capture race-associated variations and their clinical implications. To test this, we analyzed cell-cell interactions across 20 clusters using spatial proximity enrichment/depletion analysis with Giotto[35] (see Methods and Supplementary Fig. 5 for details). This analysis involved shuffling cell type labels within each ROI to establish a ROI-specific baseline of interactions, accounting for cellular heterogeneity and abundance differences, and counting interactions against the background. Differential interactions between BA and WA patients were identified using a linear mixed model (LMM), which incorporated repeated and hierarchical observations of tumor ROIs from each patient. The resulting landscape of cell-cell interactions, along with specific differential interactions, is shown in Fig. 2a, b, where positive BA/WA coefficients indicate enriched interactions and negative coefficients indicate depleted interactions in the respective race.

Our analysis revealed six BA- and five WA-specific spatial cell-cell interactions with statistical significance (Padj < 0.05) (Fig. 2b c). Although cluster abundances did not differ significantly between racial groups (Supplementary Fig. 4b), spatial interactions highlighted distinctive differences. In BA tumors, key interactions involved naïve T cells−naïve T cells, endo-mesenchymal−tumor cells, and M2 macrophages−mesenchymal cells (Fig. 2b). In contrast, WA tumors exhibited interactions in a TME, featuring helper T−helper T cells, and myeloid cell−cytotoxic cell, and T cell−cytotoxic cell interactions, with a notable presence of exhaustion marker (CD152) and hypoxia (HIF 1α) (Fig. 2c). Leave-2-patient-out cross-validation for each race confirmed the robustness of the identified interactions, with consistent coefficients and significance across all subsamples (see 95% CI in Fig. 2b, c). Importantly, when the evaluation was confined solely to the low-grade tumors, highly similar BA/WA coefficients and sets of prioritized spatial cell-cell interactions were obtained, validating the LMM modeling (Supplementary Fig. 6).

Figure 2d illustrates the spatial distribution of the top-ranked BA and WA cell-cell interactions across patient samples, with BA tumors showing enrichment in BA-specific interactions. In BA tumors, interactions between M2 macrophages (orange), endothelial cells (black), tumor cells in an epithelial-mesenchymal transition (EMT) state (green), and naïve T cells (blue) are highly clustered. These clustered interactions are not observed in WA tumors. In contrast, WA tumors display increased WA-specific interactions (Fig. 2e), such as interactions between cytotoxic cells in the hypoxic TME (orange) and exhausted T cells (pink), which are more dispersed compared to the clustered interactions seen in BA tumors. Multiplex immunofluorescence staining in BA TNBC confirmed the endothelial-macrophage interaction (CD31-CD163, Fig. 2f). Co-localization analysis of CD31 and CD163 (Fig. 2g) revealed significantly higher co-localization in BA TNBC (*n* = 27 patients) compared to WA TNBC (*n* = 44 patients), even after adjusting for cell numbers per image (Fig. 2h). These findings further validate the spatial cell-cell interactions identified by IMC.

### Racially distinct cellular interactions also exist in other IMC cohorts

To investigate whether similar racially distinct cell-cell interactions are present in other patient cohorts, we performed IMC experiments on TMAs from two additional TNBC patient groups: Roswell Park (see Supplementary Table 2 for clinical data, 46 patients) and Baylor Scott and White (BSW2, 10 patients), referred to as validation cohorts. We applied strict quality controls, excluding cores with low cell counts, limited tumor content, or partial cores. Next, we performed cell

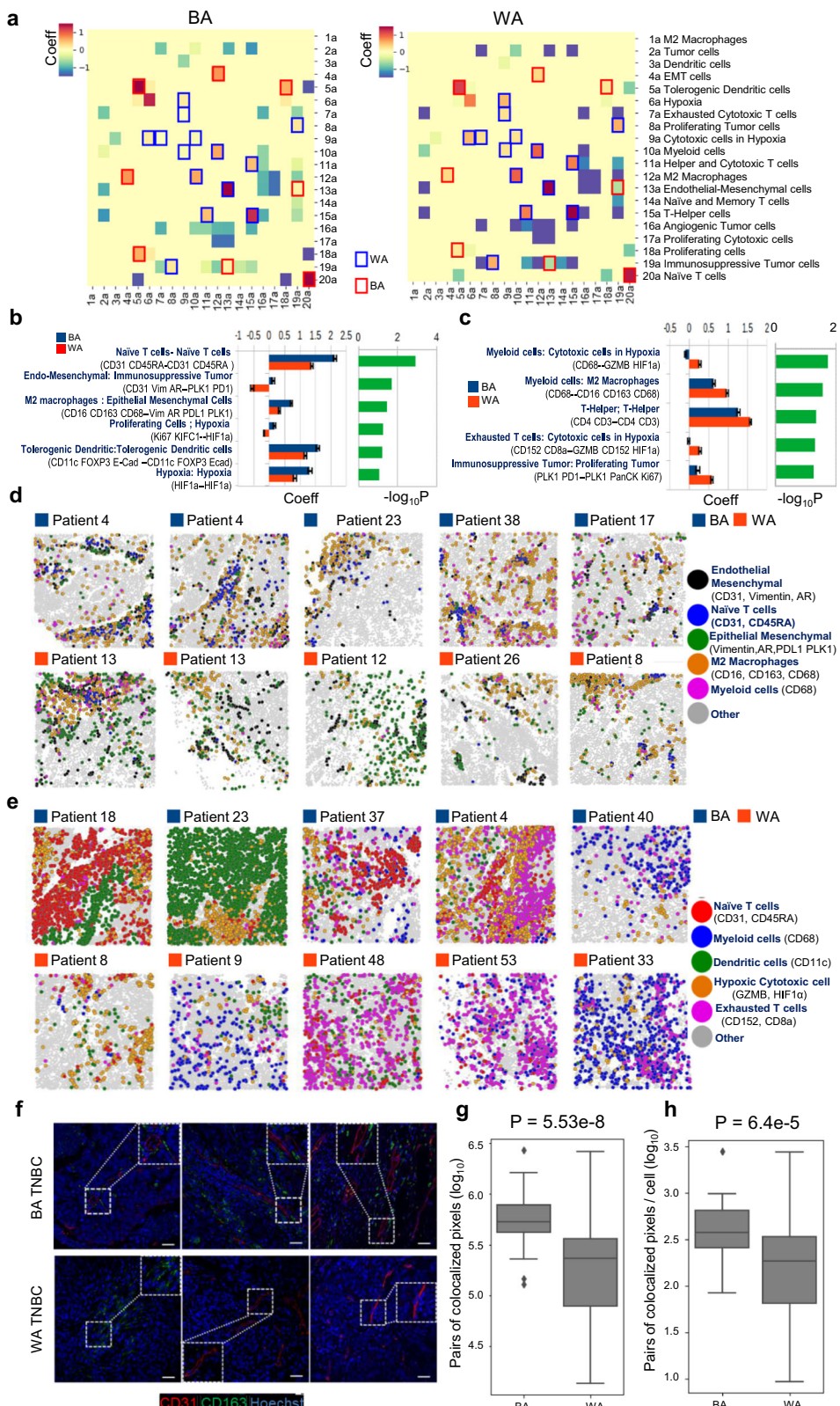

segmentation, unsupervised single-cell clustering, and spatial proximity analyses to identify BA- and WA-specific cell interactions. Each cohort yielded 20 distinct cell-type clusters, labeled with suffixes "b" (Roswell Park; Supplementary Fig. 7a) or "c" (BSW2; Supplementary Fig. 8a), respectively.

In Roswell Park, differential BA-specific cell interactions were identified, revealing interactions involved myeloid cells, endothelial cells, proliferating tumor cells, and naïve T cells (Supplementary Fig. 7b). Similarly, in BSW2, significant BA interactions featured cells undergoing epithelial-mesenchymal transition (EMT), endothelial cells, macrophages, and naïve T cells (Supplementary Fig. 8b), validating findings from the BSW discovery cohort (Fig. 2b).

For the WA group, Roswell Park showed WA interactions among exhausted T cells, Tregs, memory T cells, and M2 macrophages within

**Fig. 2 | Spatial Proximity Analysis Uncovers Distinct Cell-Cell Interactions in Black American (BA) and White American (WA) Triple Negative Breast Cancer (TNBC) Using Imaging Mass Cytometry (IMC). a** Landscape of cell-cell interactions in BA (left) and WA (right) TNBC, based on pairwise interactions between the 20 cell clusters defined in Fig. 1d. Race-specific interactions were identified using a linear mixed model, where the interaction score is modeled as score ~ race + patient, with race as a fixed effect and patient as a random effect. Coefficients for the BA (left) and WA (right) terms are shown, with positive coefficients indicating enriched interactions and negative coefficients indicating depleted interactions. Significant BA-specific interactions are highlighted with red boxes, while significant WA-specific interactions are highlighted with blue boxes. **b** Top six spatially resolved cell-cell interactions in BA TNBC (blue bars) compared to WA TNBC (red bars), with significance determined by -log10 *P* values (green bars). *P*-values are derived from 1-sided F-test, adjusted for multiple comparisons. BA TNBC tumors are characterized by Endothelial-Macrophage-Mesenchymal (Endo-Mac-Vim) interactions. Error bars represent the 95% confidence interval (CI) derived from 100 leave-2-patient-out subsampling. **c** Same as panel (**b**) but showing the top five spatially resolved cell-cell interactions in WA TNBC (red bars) compared to BA TNBC (blue bars), with significance computed by -log10 *P*-values (green bars). *P*-values are derived from 1-sided F-test, adjusted for multiple comparisons. WA TNBC tumors are marked by immune exhaustion-related interactions. Error bars represent the 95% CI derived from 100 leave-2-patient-out subsampling. **d** Spatial illustration of key BA-associated cell-cell interactions identified by IMC, showing co-localization of Endothelial (black) and Macrophage (orange) clusters in BA patients. **e** Spatial illustration of key WA-associated cell-cell interactions identified by IMC, highlighting the prominent interactions between Cytotoxic cells (orange) and Exhausted T cells (pink) in WA tumors, suggesting immune exhaustion. **f** Multiplex immunofluorescence validation of Endothelial (CD31)-Macrophage (CD163) interactions in BA TNBC, but not in WA TNBC. Representative images from three BA and three WA TNBC patients are shown. Magnified images (from the large white box) of areas marked by small white boxes are included as insets. Scale Bar: 40 um. **g, h** Pixel-level co-localization quantification of Endothelial-Macrophage interaction, shown in panel (**f**), across the entire tumor tissue (**g**) and per cell within the tumor (**h**) in BA and WA TNBC. In each boxplot, $N = 40$ images were used for the BA group (27 patients) and $N = 63$ images were used for the WA group (44 patients). *P*-values for panels (**g**) and (**h**) were calculated using the Mann-Whitney test. Box plots represent the median (center line), interquartile range (25–75%; bounds of the box), and whiskers extending to the 1.5 IQRs. Points that fall outside this range are displayed independently. All data points are used; no outlier exclusion was applied. Source data are provided as a Source Data file.

a hypoxic TME (Supplementary Fig. 7c). BSW2 demonstrated WA-specific interactions involving exhausted T cells, helper T cells, dendritic cells, and M2 macrophages (Supplementary Fig. 8c). These are consistent with the exhausted T cell phenotype in the BSW discovery cohort (Fig. 2c). Importantly, despite differences in cluster definitions across the three IMC datasets due to unsupervised clustering, a consistent set of cell types emerged across each race within the respective cohorts. Therefore, we next sought to determine if these recurrent interactions formed higher-order cellular communities and if these communities had prognostic implications.

## Cellular interactions analyzed in multiple IMC cohorts reveal existence of recurrent cell community structure that further associate with clinical outcomes in each cohort

To investigate whether cell-cell interactions extend beyond pairwise relationships to form organized cellular communities within the TME, we expanded the list of differential interactions (with FDR = 0.20) for each cohort and performed community detection. This involved identifying connected components followed by pruning (see "Methods"). As the number of interactions increased, distinct communities began to emerge, showing a high degree of consistency in the BA- and WA-specific communities across the three independent IMC datasets (BSW discovery, BSW2, and Roswell Park validation). A total of two and four communities, respectively, were identified for the BA and WA groups (Fig. 3a, b and Supplementary Fig. 9). The dominant community in BA (termed BA-Community-1) was characterized by recurring interactions involving "M2 Macrophages-Endothelial Cells-Mesenchymal Cells-Naïve T Cells" (Fig. 3a). In contrast, the top community in WA (termed WA-Community-1) was composed of "Exhausted Cytotoxic CD8 T Cells-Helper T Cells-M2 Macrophages-Hypoxia" (Fig. 3b).

We subsequently assessed the prognostic value of the BA- and WA-specific communities by examining their prevalence within the ROIs and correlating these with overall survival (OS) in patients. As previously reported, race did not significantly influence OS between the two patient groups in both the BSW Discovery and Roswell Park (Tables 1 and 2, Race). To evaluate the collective impact of the BA and WA communities, we computed a composite interaction score for the communities by initially aggregating the individual cell-cell interaction scores within each BA and WA community for each patient sample, then summed across all identified BA or WA communities to form a BA or WA-community-all score per patient. Patients were stratified into high- and low-score groups based on the average community-all score across patients. The association of

these scores with OS was examined using Kaplan-Meier (KM) curves in both the BSW-Discovery and Roswell-Validation IMC cohorts. Notably, after adjusting for age, race, and stage, higher BA-community scores were linked to worse OS in both the BSW-Discovery and Roswell-Validation cohorts (Fig. 3c, d). In contrast, the WA-community score was only prognostic in the Roswell Park-Validation cohort (Fig. 3d). We next assessed the prognostic value of BA- and WA-associated communities in each race group. BA-associated communities were a significant predictor of OS for BA patients across both the above-mentioned cohorts (Supplementary Fig. 10a, c). WA-communities were a significant predictor of OS for the WA patients in both the cohorts (Supplementary Fig. 10b, d). Specifically, the prognostic value of the WA-communities were more pronounced for WA vs BA TNBC patients (Supplementary Fig. 10 b, d). Furthermore, combining the BA and WA-interaction scores was no longer prognostic, highlighting the unique prognostic attributes of BA and WA-associated interaction scores (Supplementary Fig. 11a, b).

Consistent with these findings, in the multivariate analysis, the BA-associated community remained significantly associated with OS even after adjusting for age, race, and stage in both datasets (Tables 1 and 2). In the BSW-Discovery cohort, the BA-associated community also maintained significance when adjusted for BMI (Table 1, $P = 0.021$). Conversely, the WA-associated community was significant in the multivariate analysis only in the Roswell Park-Validation cohort (Table 2). However, none of the single-cell clusters were prognostic in BA or WA patients in the BSW Discovery data (Supplementary Table 3). Therefore, we conclude that single-cell abundances do not influence race-specific survival outcomes. In contrast, Fig. 3 in our manuscript highlights the association between communities, made of interacting cell clusters, and overall survival in both a race-dependent and independent manner.

Taken together, our analysis demonstrates that the BA-associated community is a robust prognostic marker for overall survival across multiple patient cohorts, independent of clinical factors such as age, race, and stage. This highlights the potential of cell-cell interactions within the BA community as a key determinant of tumor progression. In contrast, the WA-associated community's prognostic value appears to be cohort-specific, warranting further investigation into its role in specific patient populations. These findings underscore the importance of cellular interactions in shaping the TME and suggest that targeting community-specific interactions may offer promising therapeutic avenues for improving patient outcomes.

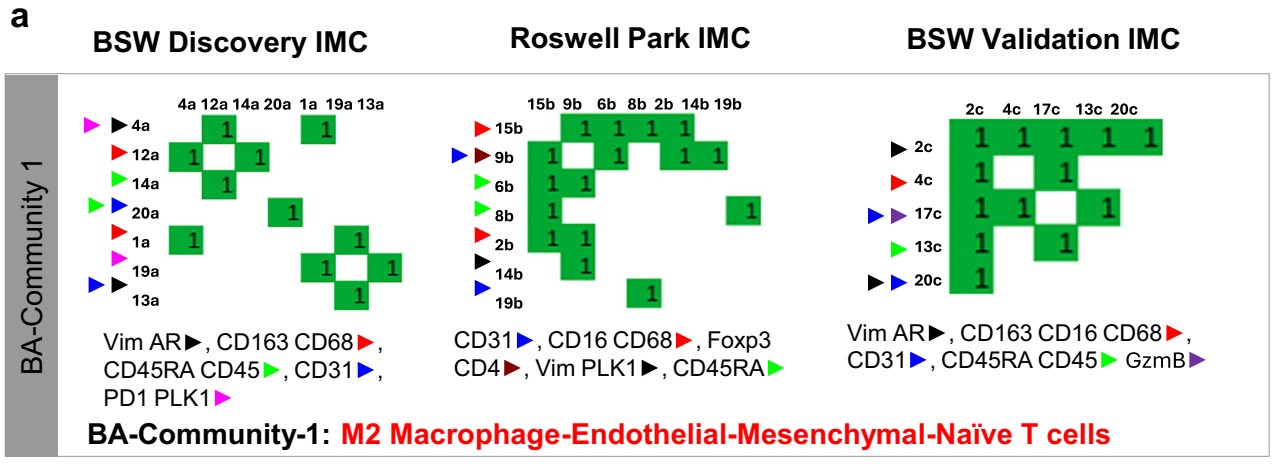

**BA-Community-1: M2 Macrophage-Endothelial-Mesenchymal-Naïve T cells**

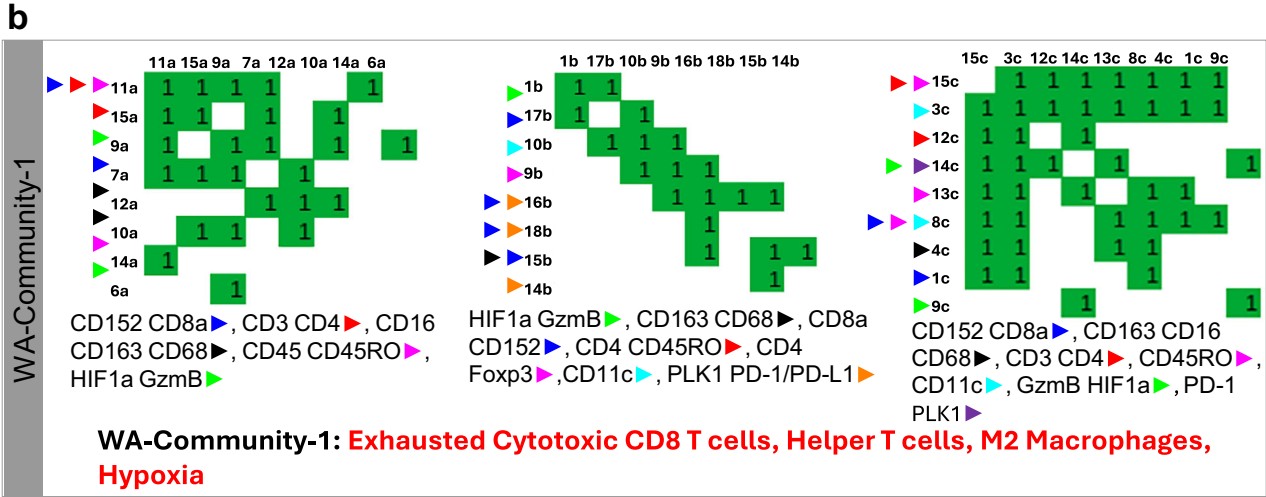

**WA-Community-1: Exhausted Cytotoxic CD8 T cells, Helper T cells, M2 Macrophages, Hypoxia**

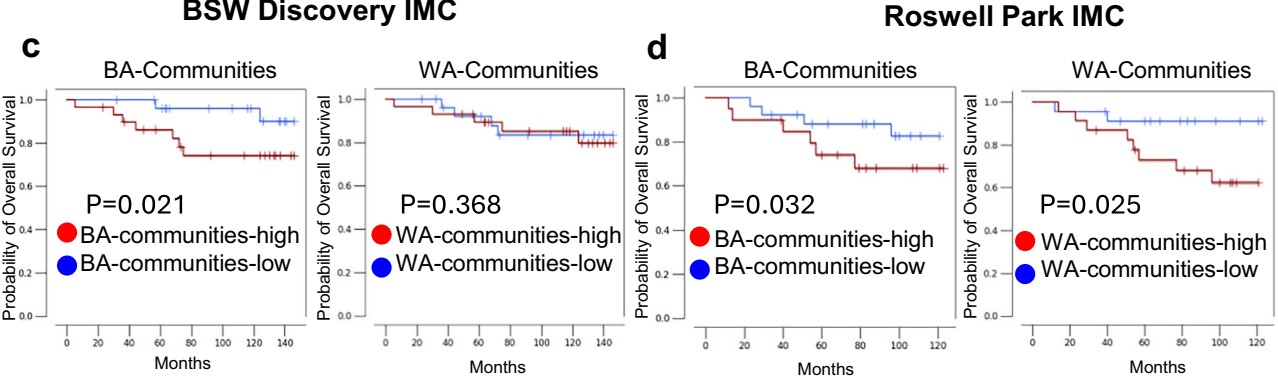

**Fig. 3 | Identification of Recurrent Cellular Communities and Their Association with Clinical Outcomes in Black American (BA) and White American (WA) Triple Negative Breast Cancer (TNBC) Cohorts. a** BA-Community-1: In the BA cohort, community detection of cellular interactions, derived from expanding differential interactions (FDR = 0.20) and identifying connected components, revealed a dominant community (BA-Community-1) characterized by recurring interactions between "M2 Macrophages-Endothelial Cells-Mesenchymal Cells-Naïve T Cells." Each heatmap shows the cell-type interactions between specific cell-type clusters (indicated in row and column labels) in each IMC dataset. A community is defined as a group of connected cell-type interactions. So, for example, BA-Community-1 in Roswell Park IMC is made of connecting interactions between clusters 15, 9, 6, 8, 2, 14, and 19b. Members of the community are indicated at the bottom of each community. This BA-community-1 was consistently observed across the Baylor Scott & White (BSW) discovery, BSW2, and Roswell Park datasets

and was associated with clinical outcomes in BA TNBC. **b** WA-Community-1: Using the approach described in (**a**), in the WA cohort, the dominant community (WA-Community-1) comprised "Exhausted Cytotoxic T Cells-Helper T Cells-Macrophages-Hypoxia." This community was consistently found across the BSW discovery, BSW2, and Roswell Park datasets. For additional detected communities in BA and WA, see Supplementary Fig. 9. **c** Kaplan Meier (KM) plot showing significantly poor overall survival among TNBC patients when patients were stratified by the sum of scores of all BA communities (1&2) after adjusting for age, race and stage, in the Imaging mass cytometry (IMC) profiles of BSW-discovery and Roswell Park validation cohort. **d** KM plot showing the stratification of TNBC patients based on the sum of scores of all WA communities (1–4) after adjusting for age, race and stage, in the IMC profiles of BSW-discovery and Roswell Park validation cohort. Stratification into the high and low groups was performed using the average. All P-values computed using a one-tailed log-rank test.

**Table 2 | Cox Proportional Hazard Analysis for the Roswell Park Validation Data.** *P*-values: Wald test, 1-sided

| Cox-proportional Hazard Ratios (for Roswell Park Cohort) | | |
|---|---|---|
| | **Hazard Ratio** | ***P*-value** |
| **Univariate model** | | |
| stage | 1.75 | 0.125 |
| age | 1.03 | 0.061 |
| race | 1.61 | 0.264 |
| BA-communities-all | 2.15 | 0.113 |
| WA-communities-all | 4.16 | 0.036 |
| **Multivariate model, adjusting for stage, age, race** | | |
| BA-communities-all | 3.75 | 0.032 |
| WA-communities-all | 6.11 | 0.025 |

### Tumor microenvironment (TME) architecture of BA tumors is composed of endothelial cells, macrophages, and tumors in a mesenchymal state

To extend our analysis beyond the limited coverage of protein markers examined with IMC, we integrated spatial transcriptomic data to gain a more comprehensive understanding of the distinct multicellular niches in BA and WA TNBC. We combined our IMC results with a publicly available, racially balanced spatial transcriptomic dataset from 20 TNBC patients (*n* = 10 each for BA and WA) derived from flash-frozen tumor samples[30]. Genes encoding proteins associated with the top BA and WA communities identified through IMC were interrogated within this spatial transcriptomic dataset to assess their spatial localization across the 20 TNBC tumors (Fig. 4a, b). To start, protein markers identified by IMC, such as CD31 and CD45RA, were converted to their corresponding genes, *PECAM1* and *PTPRC*, in the spatial transcriptomic dataset. Then, the genes belonging to each BA-Community 1 or WA-Community 1 (Fig. 4b) were spatially plotted to reveal their spatial localization patterns in tumors.

Notably, the localized regions demonstrating the co-expression of BA-Community-1 genes (characterizing the Endo-Mesenchymal-Mac-Naïve T niche) are significantly more spatially clustered in BA patients compared to WA patients (Fig. 4c, black outlines, and spatial clustering score Fig. 4d). Strong spatial clustering observed across 10 BA TNBC patients indicates that this is a recurrent feature in BA tumors. Conversely, WA patients exhibited a random and uniform distribution for these genes (Fig. 4c, WA), suggesting that endothelial markers and vimentin (mesenchymal marker) are either not co-expressed or expressed at diminished levels in WA patients. Taken together, these findings imply that the mesenchymal cell, endothelial cell, and macrophage form a multi-member multicellular TME niche that is distinctly recurrent across BA TNBC.

In contrast to our observations with the BA query, the localized spots for the WA-Community-1 query genes are uniformly distributed throughout the entire tumors (Supplementary Fig. 12) rather than confined to any region, suggesting an infiltration of exhausted T cells throughout the tumors. Instead, WA tumors were characterized by a higher presence of WA-Community-1 spots, irrespective of their spatial configuration (Supplementary Fig. 12). In summary, WA interactions are chiefly characterized by exhaustive immune and hypoxic environments (illustrated by the diffused co-localization of markers *GZMB, CTLA4, HIF1A*) in WA TNBC.

### Niche-specific differential expression analysis reveals additional players associated with BA and WA-tumor associated multicellular niches

To further delineate molecular factors underlying race-associated communities, we extracted ST spots that exhibit high and low expression for each community and performed niche-specific gene signature analysis (refer "Methods"). In this context, spots characterized by elevated expression of BA- or WA-community genes were compared with spots exhibiting low expression within the respective ST samples, to identify co-expressed genes called extended signature genes (ESGs). A differential analysis of ESGs between BA and WA tumors further identified race-associated ESGs that could suggest additional cell types in the niches (Fig. 5a). With these ESGs, we also observed a higher number of spots localized to each respective race-associated community (Fig. 5d).

As illustrated in Fig. 5b, ESGs associated with the BA-Community-1 query implicated endothelial cells and macrophages, and additionally Cancer Associated Fibroblasts (CAFs). The cell-type specific expression profiles reveal that the expression of the ESG genes are markedly higher in BA than WA tumors within these specific cell type compartments (Fig. 5b). Notably, CAFs are marked by the higher expression of *SPARC, COL4A1, TAGLN, CALD1, FBLN1, RARRES2, CCDC80, SFRP2*, and *VCAN* in BA patients. Intriguingly, BA-niche ESGs lack markers of B-cells, T-cells, neutrophils, and epithelial cells. Gene set enrichment analysis (GSEA) of the BA-associated ESGs revealed a strong association with processes such as extracellular matrix organization (*P* = 2.3E-28), epithelial-mesenchymal transition (EMT, *P* = 6.6E-28), tumor vasculature development (*P* = 1.2E-46), and endothelial cell activity (*P* = 1E-18) in BA TNBC (Fig. 5e).

In contrast, the ESGs associated with the WA-Community 1 (Fig. 5c) included genes that characterize T cells and additionally neutrophils. GSEA of the WA-associated ESGs highlighted alterations in immune signaling pathways within WA TNBC (Fig. 5f). These ESGs included *S100A8* and *S100A9*, which are well-established markers for neutrophils, as well as *CD3E, IL2RG*, and *LTB*, which are indicative of T cell populations. Importantly, the WA-niche ESGs did not include markers for CAFs or endothelial cells (Fig. 5c). Notably, neutrophils were not evaluated in the initial BSW-discovery IMC panel, as it was devoid of any neutrophil marker; however, it was identified by this ST-derived ESG analysis. Therefore, to corroborate these results, in formulating the IMC antibody panel for the BSW2 and Roswell validation cohorts, we incorporated the neutrophil marker Myeloperoxidase (MPO) to measure its expression. Our results showed that the expression of *MPO* strongly correlated with WA-Community-1. Specifically, *MPO* expression was highly enriched in cluster 9c of the BSW2 dataset and cluster 1b of the Roswell Park IMC dataset (Supplementary Fig. 13a, b). The interaction between neutrophils and exhausted T cells is more pronounced in WA vs BA TNBC (Supplementary Fig. 13c, d). These findings confirm the presence of neutrophils within the top WA community and confirm the approach of using ESGs to identify additional niche constituents.

### Drivers in BA and WA tumor niches, and the predominance of exhausted T cells in WA niches

To investigate the drivers within the racially distinct spatial environments, we used ESGs, as described earlier, to extract and reveal a unique set of ligand-receptor pairs within the BA and WA niches (Fig. 6a). In the BA niche, most ligand-receptor interactions were linked to processes such as cell-cell communication, platelet-derived growth factor (*PDGF*) signaling, epithelial-mesenchymal transition (EMT, TGF-beta), Wnt/Notch signaling, endothelial/extracellular matrix remodeling, vascular endothelial growth factor (*VEGF*) signaling, and integrin-mediated signaling. These interactions primarily involved perivascular-like cells, endothelial cells, CAFs, and macrophages (Fig. 6b). In contrast, the ligand-receptor interactions in the WA niche were primarily associated with immune cells, as supported by the normalized frequencies of the interactions between neutrophils, T cells, and myeloid cells (Fig. 6c). Notably, inflammatory cytokines such as *CXCL9*, and *CXCL11* showed prominent interactions with the *CXCR3* receptor. These ligand-receptor interactions may have

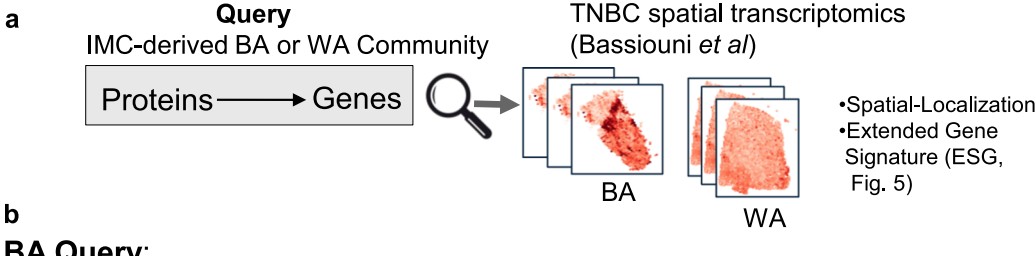

**a**

**Query**
IMC-derived BA or WA Community

Proteins ⟶ Genes

**TNBC spatial transcriptomics**
(Bassiouni *et al*)

- Spatial-Localization
- Extended Gene Signature (ESG, Fig. 5)

BA

WA

**b**

## BA Query:

**BA-Community1:** M2 Macrophage-Endothelial-Mesenchymal-Naïve T cells
**Protein***: CD31 CD45RA Vim AR CD16 CD68 CD163
**Gene:** *PECAM1 PTPRC VIM AR FCGR3A FCGR3B CD68 CD163*

## WA Query:

**WA-Community1:** Exhausted Cytotoxic CD8 T cells, Helper T cells, M2 Macrophages, Hypoxia
**Protein:** CD152 CD8a CD4 CD163 CD68 CD45RO HIF1a GzmB PLK1 PD-1 CD3
**Gene:** *CTLA4 CD8A CD4 CD163 CD68 PTPRC HIF1A GZMB PLK1 PDCD1 CD3E CD3D CD3G*

**c** **Spatial Localization (Niche)** **d**

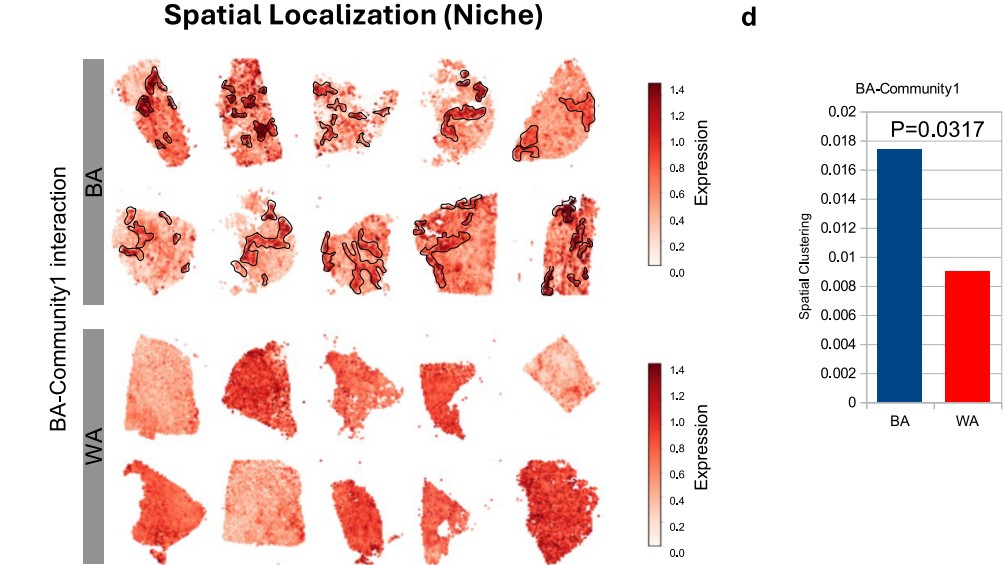

**Fig. 4 | Black American (BA)-Associated Community Exhibits Unique Spatial Clustering in Triple Negative Breast Cancer (TNBC) Spatial Transcriptomics Data. a** Overview of the approach used to identify the spatial localization of BA- and White American (WA) -associated communities in TNBC tissues. BA or WA-associated communities, defined by IMC-derived protein markers of interacting cell types, were converted to gene names and interrogated in a racially balanced TNBC spatial transcriptomics (ST) dataset to assess spatial patterns of co-localization for the query genes. **b** The top BA-associated community (BA-community 1) was interrogated in the ST dataset[30]. Protein symbols (blue) and corresponding gene symbols (black) are shown. **c** Representative example of spatial co-localization for BA-community 1 in 10 BA and 10 WA TNBC samples from the ST dataset. BA-community 1 genes show strong co-localization (marked by black outlines) in BA TNBC, but not in WA TNBC. The scale bar represents the strength of co-localization, defined as the averaged log-transformed normalized read count. **d** Quantification of spatial clustering of co-localized spots for BA-community 1 in BA (blue) and WA (red) TNBC tumors. Each bar represents the average of 10 BA and 10 WA samples shown in panel (**c**). Spatial clustering of BA-community 1 was significantly higher in BA TNBC compared to WA TNBC, with statistical significance determined by the T-test. Source data are provided as a Source Data file.

downstream consequences on the expression of target genes and activities of signaling pathways.

To further explore the functional state of T cells within the WA niches, we examined the expression of a T-cell exhaustion signature in the ESGs associated with the WA niche. As shown in Fig. 6d, the WA-associated niche (WA-Community-1) exhibited the presence of an 11-gene T-cell exhaustion signature, including *TRBC2*[36]*, LAG3*[37]*, HAVCR2*[38]*, and CSF1*[39]. The number of spots expressing these exhaustion genes was significantly higher in WA tumors compared to BA tumors (Fig. 6e). Collectively, these results underscore the unique TME features in BA and WA TNBC, with the WA TME exhibiting a signature characterized by exhausted T cells, along with neutrophils and myeloid cells, contributing to a distinct immune landscape.

## Validation of BA and WA-ESG signatures in independent clinical cohorts

We then validated the BA and WA-associated ESGs described in Fig. 5 by employing a combination of 10X Visium ST and Nanostring GeoMx DSP systems. As previously outlined in Fig. 1, tumor microarrays (TMAs) were created and Visium ST profiling was conducted on two independent cohorts: BSW2 (9 samples, 5 BA and 4 WA, Supplementary Figs. 14 and 15 for H&E images) and Georgia-Validation (10 samples, 6 BA and 4 WA, Supplementary Figs. 16 and 17 for H&E images). Using both cohorts, we confirmed the significantly higher expression of BA-ESGs, which encode endothelial-macrophage-EMT interactions, in BA tumors, and WA-ESGs, which encode exhausted T cells and neutrophils, in WA tumors (Fig. 7a for the BSW2 cohort, and Fig. 7b for

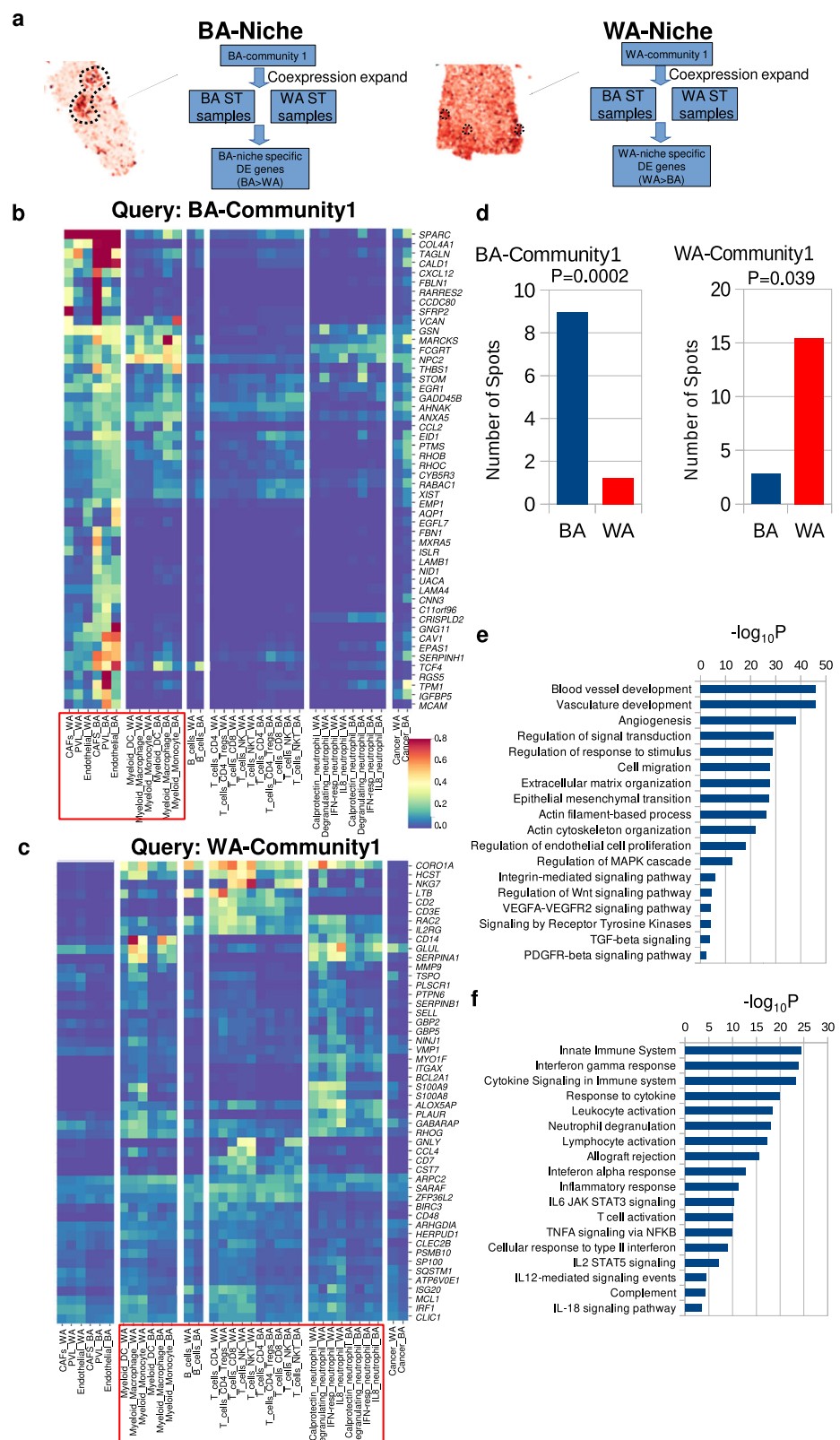

the Georgia cohort, highlighting the top BA-Community-1 and WA-Community-1). In addition, through spatial localization, we observed that BA-community genes spatially clustered strongly within and between themselves in BA tumors, whereas no such clustering was seen in WA tumors in both validation cohorts (Fig. 7c).

Nanostring GeoMx evaluation of 26 BA and 31 WA TNBC tumors, employed for imaging mass cytometry in Fig. 1 (refer to

Supplementary Fig. 2 for indicated regions of interest), further substantiated the increased expression of ESGs linked with the BA niche in BA tumors and the WA niche in WA TNBC (Fig. 7d). Importantly, the immune compartment (CD45-positive) demonstrated a notable increase in ESG expression (Fig. 7d), while the epithelial compartment (PanCK-positive) did not exhibit such an increase (Supplementary Fig. 18), highlighting the compartment-specific characteristics of the

**Fig. 5 | Distinct Gene Signatures Define Unique Cell Types in Black American (BA) and White American (WA) Tumor Niches. a** Overview of the approach used to identify niche-associated gene signatures. For BA- and WA-associated communities (BA-community 1 and WA-community 1), extended signature genes (ESGs) were derived based on co-expression within the query (i.e., BA-community 1 or WA-community 1 genes), co-localized spots. ESGs were computed per sample and compared between BA and WA, with genes significantly elevated in BA (i.e., BA > WA for BA-niche) or in WA (i.e., WA > BA for WA-niche) selected for further analysis. Each query generated a distinct BA-niche and WA-niche, composing of ST spots uniquely expressing BA-community 1 or WA-community 1 genes. **b** Representative heat map showing cell-type-specific expression of ESGs associated with the BA niche defined by expansion of BA-community 1. These ESGs correspond to various cell types, including Cancer-Associated Fibroblasts (CAF), endothelial cells, and myeloid cells, while the BA-niche is poor in T-cells and neutrophils. **c** Same as in (**b**), but for the heat map showing cell-type-specific expression of ESGs associated with the WA niche defined by expansion of WA-community 1. ESGs in this niche predominantly correspond to exhausted T-cells and neutrophils, while the WA-niche is poor in CAFs and endothelial cells. **d** Average quantification of the number of co-localized Visium spots per sample across 10 BA (blue) and 10 WA (red) TNBC samples corresponding to BA- and WA-associated niches. Statistical significance was computed using a T-test. **e, f** Gene set enrichment analysis (GSEA) of ESGs associated with the BA-niche (**e**) and WA-niche (**f**), with -log10 P-values shown. P-values were derived from a 1-sided hypergeometric test, adjusted using g:SCS method (g:Profiler). Source data are provided as a Source Data file.

BA and WA cellular communities. Collectively, these findings offer insights into the distinct spatial ecosystems underlying WA-specific immunosuppression.

## Discussion

In this study, we introduce an integrated multi-omics analysis examining the distinct tumor microenvironments (TMEs) of BA and WA patients with TNBC. Using advanced spatial technologies like IMC and spatial transcriptomics, we identified race-associated spatially resolved cell–cell interactions, community structures, and niche signatures (i.e., ESGs) in BA and WA TNBC tumors. Importantly, the robustness and generalizability of these observations were reinforced through reproducibility across three independent IMC datasets (over 100 patients), confirming the distinct TMEs are not cohort-specific. In addition, validation in two independent cohorts, BSW2 and Georgia, using complementary 10X Visium and Nanostring GeoMx Digital Spatial Profiling (DSP) technologies further substantiated our findings.

Our study builds upon recent studies that have highlighted differences in TNBC between BA and WA patients[16,20,29,40–44], which frequently relied on bulk transcriptomics and proteomics analyses. While these studies uncovered distinct molecular features between these groups, they often lacked insights at the single-cell level. Specifically, the bulk approach cannot resolve the complexities of the TME, nor can it account for the spatial relationships between various cell types. Our study addresses this gap by using single-cell spatial proteomics alongside transcriptomics to map the cellular architecture and interactions within the TME of both BA and WA TNBC, revealing how these spatially organized cell communities may drive tumor progression and influence clinical outcomes.

Focusing on the biological insights derived from these spatial patterns, our spatial analysis identified distinct racial differences primarily in the spatial arrangement and interaction patterns of cell types, even though the overall abundances of major cell clusters were comparable between BA and WA patients. BA tumors showed tightly clustered, immune-suppressive communities involving naïve T cells, dendritic cells, and M2 macrophages, often located within hypoxic regions; these structures appear capable of promoting immune tolerance, EMT, and potentially metastasis[45] through localized signaling interactions. Conversely, WA tumors displayed communities characterized by interactions of exhausted T cells and myeloid cells, indicative of chronic immune suppression. This type of TME may hinder effective anti-tumor immunity and contribute to poor treatment response. Overall, our study emphasizes that the spatial organization of cellular communities and their interactions in the TME, rather than just cell composition, are key drivers of racial disparities in TNBC.

While IMC-derived single-cell interactions provide valuable biological insights, its capacity is limited by the number of measurable markers, restricting the range of cell–cell interactions that can be examined. To overcome this constraint, we utilized IMC-identified protein markers indicative of race-associated communities as "queries" and integrated them with spatial transcriptomic data. This integration allowed us to reconstruct the multicellular niche structure,

architecture, and composition in BA and WA TNBC tumors more comprehensively. This strategy enabled us to extend the scope of race-associated cell-cell interactions and gain deeper insights into other relevant cell types and their interactions within the TME. Through this approach, we identified race-associated ESGs, which uncovered previously unrecognized aspects of the TME in both BA and WA tumors—insights that were not apparent through prior global clustering methods. These findings align with earlier research emphasizing the advantages of niche-specific analysis for understanding complex tumor ecosystems[46].

Analysis of the integrated data revealed distinct ESG signatures that further elucidated the biological differences and potential therapeutic vulnerabilities between the racial groups. The BA-niche characteristics emphasized the significant role of cancer-associated fibroblasts (CAFs) and enrichment in pathways promoting EMT, vascular development, TGF-β, Wnt/Notch1 signaling, and extracellular matrix (ECM) organization. These signatures indicated that BA tumors had niches rich in macrophages, accompanied by suppressed CD8 + T cell function. This phenotype aligns with known mechanisms of immune suppression, including that driven by TGF-β through ECM remodeling[47], and metabolic reprogramming and reduction in arginine, along with the secretion of ornithine by tumor-associated macrophages[47] that suppress T cell functions. In addition, the presence of vascular development genes (*VEGF* and thrombospondin[48]) within BA-associated ESGs suggests potential resistance mechanisms to anti-PD1 treatments, a finding requiring further investigation. Notably, similar findings in a metastatic melanoma study identified angiogenesis-related genes as contributors to immunotherapy resistance[49,50]. Conversely, WA-associated ESGs showed enrichment of exhausted T cell and neutrophil markers, paralleling patterns observed in chemotherapy-induced immune suppression in ovarian cancer. Specifically, chemotherapy treatment in ovarian cancer can instill a spatially exhausted T cell environment[51]. This suggests WA patients may harbor TMEs pre-disposed to certain therapy resistances, being less responsive to chemotherapy, highlighting distinct immunosuppressive mechanisms in WA TNBC.

Contextualizing these observations, our findings can be positioned within the landscape of known TNBC heterogeneity. Although TNBC lacks the expression of the three hormonal receptors, recent studies have identified additional molecular subtypes, including basal-like immune-suppressed, immunomodulatory, luminal androgen receptor (LAR), and mesenchymal-like subtypes[52]. Furthermore, spatial immune classifications have emerged; Hammerl et al.[53] found that TNBC tissues could be divided into three phenotypes: excluded, ignored, and inflamed. This latter finding builds upon the work of Gruosso et al.[54], which showed the presence of immune-cold, immune-desert, fully inflamed, stroma-restricted, and margin-restricted TNBC tumors. Applying these subtypes to our data, as shown in Supplementary Fig. 19, we suggest that WA tumors frequently resemble the inflamed spatial phenotype ($P = 8.9 \times 10^{-17}$) whereas BA tumors tend to align more closely with the Excluded ($P = 0.0009$) and Ignored phenotypes ($P = 1.6 \times 10^{-7}$) defined by Hammerl et al.[53]. This finding further

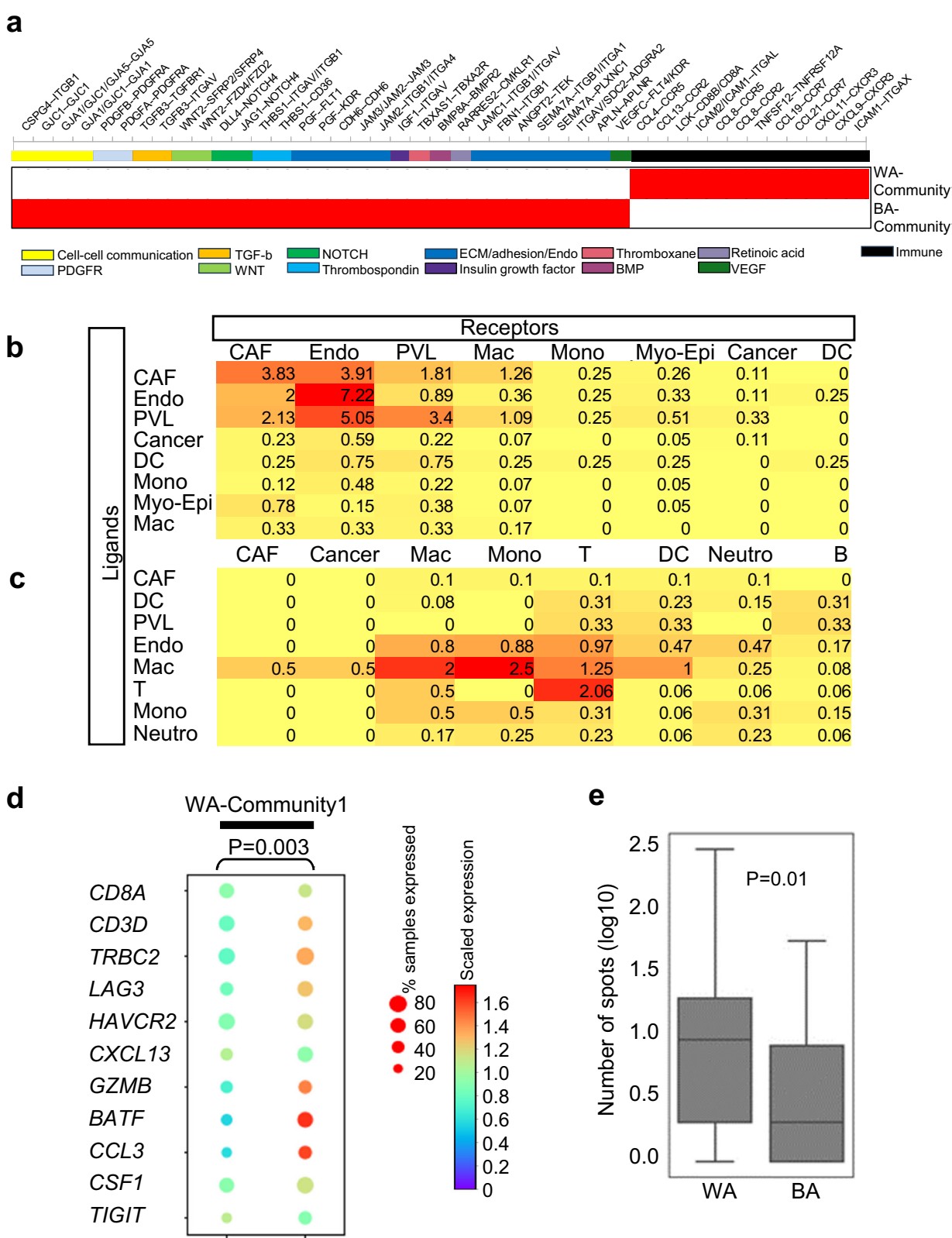

supports the notion that tumors in BA patients are more likely to be immunologically cold, and consequently resistant to immunotherapy. Consistent with this, the BA-Community-1 identified in our study overlaps significantly with the LAR and mesenchymal-like subtypes described by Jiang et al.[52]. The presence of androgen receptor expression in BA-Community-1 further supports its resemblance to the LAR subtype. In addition, the TME in BA tumors also exhibits features of the margin-restricted (showing exclusion of T cells) and immune-desert subtypes described by Gruosso et al.[54]. In contrast, WA-associated community-1 resembles the immunomodulatory subtype described by Jiang et al. and the fully inflamed subtype described by Hammerl et al.[53]. Future functional validation, potentially through

**Fig. 6 | Ligand-Receptor Interactions and ESG Analysis in Black American (BA) and White American (WA) Niches Reveal Distinct Mechanisms of Tumor Progression and Immune Suppression. a** Ligand-receptor interactions derived from ESGs associated with the BA niche (BA-community 1) reveal the activation of multiple tumor-promoting pathways, including those involving cell surface proteins, adhesion molecules, extracellular matrix components, endothelial cells, and key signaling pathways such as platelet-derived growth factor receptor (PDGFR), Transforming Growth Factor beta (TGF-ß), WNT, Notch, Thrombospondin, and Placental Growth Factor (PGF). In contrast, ligand-receptor interactions in the WA niche (WA-community 1) are dominated by cytokine activities and pathways related to T-cell exhaustion. **b** Normalized frequencies of ligand-receptor interactions in the BA-niche, analyzed using ESGs from BA-community 1, are presented in a matrix format with ligands (rows) and receptors (columns). **c** Normalized frequencies of ligand-receptor interactions in the WA-niche, analyzed using ESGs from WA-

community 1, are similarly presented. **d** Average expression of genes associated with T-cell exhaustion[31] in WA-community 1 in WA and BA TNBC tissues. The expression of T-cell exhaustion-related genes was significantly higher in WA TNBC compared to BA TNBC, with significance computed using a T-test. **e** Quantification of the average number of spots containing genes associated with T-cell exhaustion in $N = 10$ BA and $N = 10$ WA TNBC spatial transcriptomics (ST) samples[30]. Each ST sample is an independent TNBC patient and contains between 1500 and 2500 spots. The number of co-localized spots was significantly higher in WA TNBC compared to BA TNBC, with significance determined using a Kolmogorov-Smirnov test. Box plots represent the median (center line), interquartile range (25–75%; bounds of the box), and whiskers extending to the 1.5 IQRs. Points that fall outside this range are displayed independently. Source data are provided as a Source Data file.

assessment of immune signaling activity or localized cytokine profiling within these niches, is warranted to confirm these phenotypic characterizations. Nonetheless, these observations underscore the importance of spatial datasets for accurately assessing the function of immune cell types, revealing complexities that may be underestimated by cell type abundances or deconvoluted proportions from bulk RNAseq studies, especially when comparing BA and WA patients. Consequently, previous conclusions[15] regarding relative abundances of T cells or Treg cells between races and their functional implications might require reassessment using spatial technologies, and using spatial immunophenotypes such as ones described in this work and others (Gruosso et al. and Hammerl et al). Our work further expanded this knowledge by including additional cell-cell interactions and community network architectures as defining features of race-specific TNBC tumors.

Moving beyond classification, the implications of finding unique spatial niches are significant, extending to therapeutic avenues and prognosis. Our study revealed architectural differences within the TME between these groups, which may influence both tumor progression and response to treatment. These findings provide preliminary evidence supporting the potential development of therapeutic strategies targeting the endothelial-macrophage-EMT axis, which appears particularly prominent and thus may be especially beneficial for BA patients with TNBC. The identification of ligand-receptor interactions that regulate this axis—such as integrin-TGF-β[55], Wnt/Notch1, and VEGF signaling—highlights several actionable targets[55]. For example, clinical-grade Wnt/Notch1 inhibitors, which affect endothelial cells[56] and EMT[57,58], could represent a promising avenue for BA patients with TNBC. In addition, therapeutic regimens targeting macrophages have shown promise in preclinical models of TNBC[59]. The identification of a neutrophil-rich TME co-existing with exhausted T cells in WA tumors is particularly intriguing, a feature that was confirmed using myeloperoxidase as a neutrophil marker in our validation cohort. Recent studies suggest that neutrophils can promote tumor progression[60] and may do so independently of macrophages[61], exerting an immunosuppressive function[61,62]. These findings, in conjunction with previous studies[61,62], suggest that WA patients with TNBC often have inflamed tumors that have nonetheless evolved distinctive immunosuppressive mechanisms that involve neutrophils, providing further insight into resistance to current therapies.

Building on the discussion of the TME biology associated with racial differences, our findings suggest that race-specific cell-cell interactions have important implications for survival outcomes in TNBC. Crucially, our analysis indicates that these spatial interactions and community structures provide prognostic information beyond what simple cell abundance reveals. As demonstrated (Supplementary Table 3), single-cell abundance alone often lacked significant prognostic value, whereas the spatial context—reflecting the proximity required for direct signaling and localized immune modulation, particularly within the suppressive BA niches—proved significantly

associated with survival outcomes. While the presence of BA-tumor associated communities was correlated with poor OS in both BA and WA populations, the impact of WA-tumor associated communities on OS was more pronounced in WA patients. Specifically, WA-Community-1 was associated with poor OS primarily in WA patients in the Roswell cohort. We acknowledge that cohort-specific differences in prognostic associations, such as this WA-Community-1 finding, primarily in the Roswell cohort, could potentially be influenced by unmeasured factors. Within the measured ones, these associations remained significant after adjusting for key clinical factors such as age, race, and stage, highlighting the robustness of these TME features as potential prognostic indicators.

Clinically, these niche-specific signatures hold potential utility beyond prognosis; they could serve as biomarkers to stratify patients for clinical trials evaluating niche-targeting therapies (e.g., CAF or macrophage inhibitors in BA-like niches) or to predict response to existing immunotherapies based on the specific immune architecture (e.g., inflamed vs. excluded). The ability to identify specific cellular communities and their interactions within the TME offers promising avenues for understanding how these interactions influence disease progression in a race-dependent and independent manner and in shaping immunotherapy responses in breast cancer patients[13], further validating the significance of these findings for the development of personalized therapies.

It is crucial to recognize that while our study focuses on the biological and clinical aspects of race in TNBC, race itself is a social construct deeply intertwined with structural inequities. These inequities, including limited access to healthcare, lack of culturally appropriate care, poor nutrition, environmental exposures, poor built environments, and higher levels of chronic stress, disproportionately affect minority populations and contribute to poorer health outcomes. Addressing these socioeconomic factors would be an area of future investigation where such data are available. Moreover, health inequities extend beyond race and are prevalent across various groups experiencing social stratification, which compounds the challenges faced by marginalized populations. While advancements in medical treatment and prevention have reduced overall mortality rates, structural inequalities have led to a widening gap in health outcomes for disadvantaged groups[32]. Our research suggests that the unique TME observed in BA patients' tumors, characterized by endothelial-macrophage-EMT axis alterations, could represent a biological manifestation influenced by such factors and potentially be amenable to targeted therapies that are currently under investigation. By conducting racially inclusive research, we can better understand how different social determinants may shape the biology of cancer, thus paving the way for more equitable therapeutic strategies. Ultimately, addressing these inequities through comprehensive biological and socio-epidemiological research is necessary to reduce the disparities in breast cancer outcomes, ensuring that health improvements do not leave behind underserved populations.

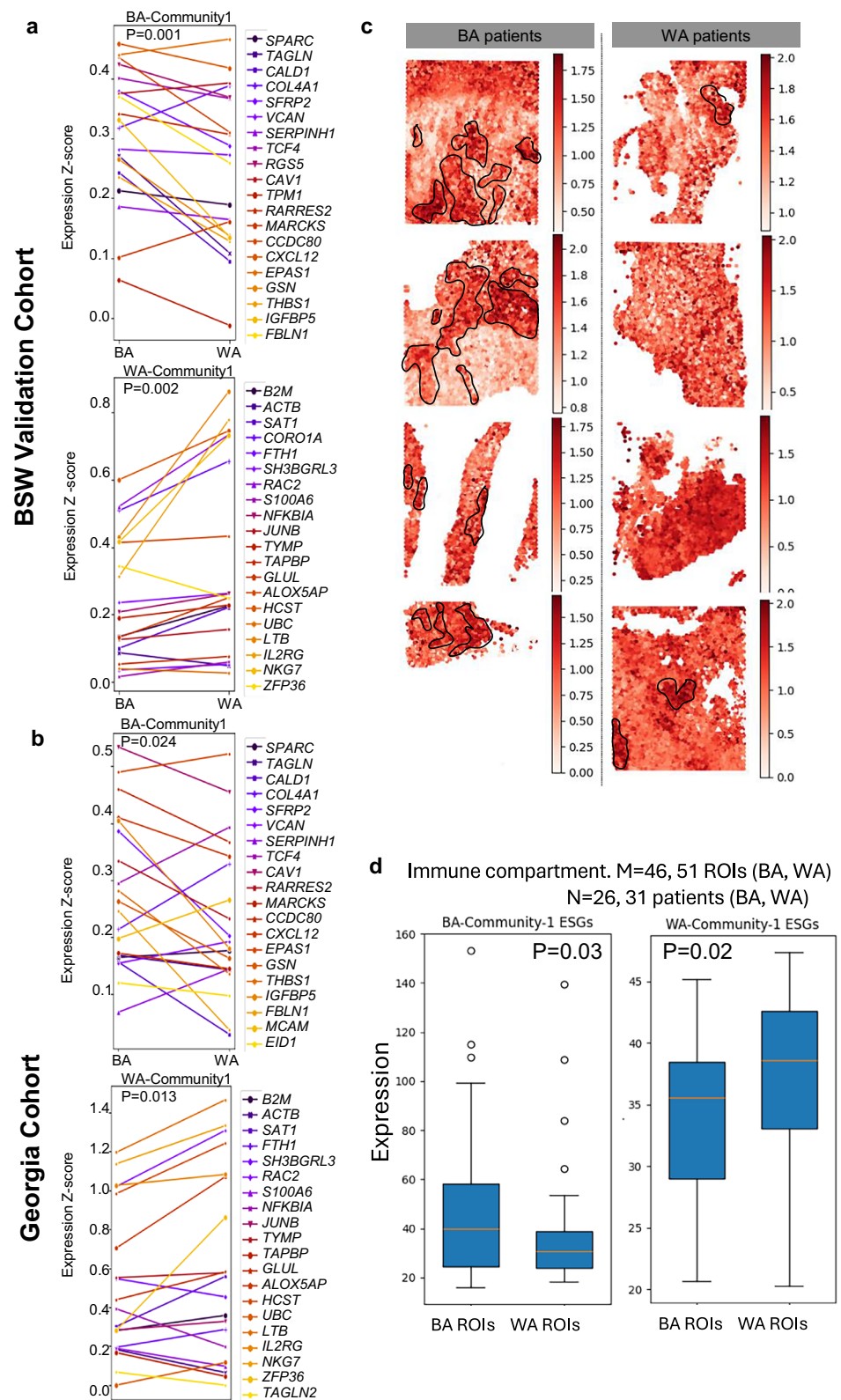

In conclusion, our study provides compelling evidence of racial differences in the spatial interactions and associated gene signatures within the TME of TNBC patients, as illustrated in Fig. 8. The characterization of the TME predominantly found in BA patients shares significant similarities with the previously described "Tumor Microenvironment of Metastasis (TMEM) doorway" model[63–66]. In the BA TME (Fig. 8a), perivascular macrophages, mesenchymal cancer cells,

and endothelial cells form a tightly connected network hypothesized to promote endothelial cell junction dissociation, thereby creating vascular openings that enable cancer cells to intravasate. These unique spatial interactions characteristic of many BA tumors are strongly correlated with poor survival outcomes in both racial groups, underscoring their clinical relevance. In contrast, the WA TME (Fig. 8b) is characterized by distinct niches enriched in exhausted T cells and

**Fig. 7 | Independent Validation of Black American (BA) and White American (WA) Niches Using Spatial Transcriptomics and GeoMX DSP System (Nano-String). a** Spatial Transcriptomics Validation in Baylor Scott & White (BSW) 2 Cohort: Top panel: Spatial transcriptomics analysis of 4 independent FFPE BA tumors and 5 WA tumors from the BSW2 validation cohort verifies the elevated expression of extended gene signatures (ESGs) associated with BA-community 1 in BA TNBC compared to WA. Bottom panel: The same analysis verifies the elevated expression of ESGs associated with WA-community 1 in WA TNBC compared to BA. *P*-values were derived from 1-sided paired *t* test. **b** Spatial Transcriptomics Validation in Georgia Cohort: Top panel: Spatial transcriptomics analysis of 6 FFPE BA tumors and 4 WA tumors from the Georgia validation cohort confirms the elevated expression of ESGs associated with BA-community 1 in BA compared to WA TNBC. Bottom panel: The same analysis verifies the elevated expression of ESGs associated with WA-community 1 in WA compared to BA TNBC. *P*-values were derived

from 1-sided paired *t* test. **c** Representative Co-localization of ESGs Associated with BA-Niche: Spatial co-localization of ESGs associated with BA-niche is observed in BA TNBC (left), but not in WA TNBC (right). Representative images of 4 tumors per group are shown. **d** GeoMX Validation of BA and WA Niche Enrichment: Community analysis of Nanostring GeoMx data from $M = 46$ BA ROIs (26 BA patients) and $M = 51$ WA ROIs (31 WA patients), which are the same TNBC tumors used for IMC in Fig. 1. Results show significantly higher enrichment of BA-community 1 (Endo-Mac-EMT) in the immune compartments (CD45 positive) of BA compared to WA TNBC. Similar analysis shows significantly higher enrichment of WA-community 1 (exhausted T cells) in the immune compartments of WA compared to BA TNBC. Significance was computed using T-tests. Box plots represent the median (center line), interquartile range (25–75%; bounds of the box), and whiskers extending to the 1.5 IQRs. Points that fall outside this range are displayed independently. Source data are provided as a Source Data file.

neutrophils. This finding raises important questions regarding the potential and immunosuppressive roles of neutrophils in this context, which require further investigation. Collectively, these findings emphasize the importance of incorporating racially diverse samples in TME research to fully appreciate the complex biological and clinical implications of racial differences in TNBC. Understanding these differences could pave the way for more personalized and effective therapeutic strategies that address the unique needs of patients from diverse racial backgrounds.

## Methods

### TNBC patient tissue microarrays
All the human studies were performed under IRB (Institutional Review Board) protocols 020-393 and 130559 (Baylor Scott and White Hospital), H-28445 (Baylor College of Medicine), H21060 (Georgia State University), 300009407 (University of Alabama at Birmingham) and Roswell Park Comprehensive Cancer Center. The approved BSW IRB protocol 020-393 waived the requirement of authorization based on 45 CFR 164.521(i)(2)(ii) and determined informed consent is not required as allowed under 45 CFR 46.116 (g). Patients from Roswell Park Comprehensive Cancer Center were consented. For patients from the Georgia cohort, the IRB H21060 from Georgia State University approved that patient consent will not be required since all samples used are archival and were de-identified to maintain patient privacy and anonymity.

De-identified breast cancer patient samples with at least 10-year follow-up and self-reported race were obtained from Baylor Scott and White Hospital (BSWH), Temple, Texas, in the form of formalin fixed paraffin embedded (FFPE) tissue. Tissue cores from 57 tumors (26 BA tumors and 31 WA tumors) spread across 8 tissue microarrays (TMAs) were used for this study. In addition to the self-reported race, clinical information such as body mass index (BMI), receptor status, tumor grade, chemotherapy status, presence of metastases, time to clinical follow up, PIK3Ca and TP53 mutation status, and information on Diabetes Miletus (DM), as well as detailed histopathology of the tumors were available to us (refer Supplementary Table 1 for clinical information and Supplementary Fig. 1 for H&E images). 2 BA and 2 WA patients received neoadjuvant chemotherapy, the remaining patients were treatment naïve at the time of sample collection. All tumors were collected at surgery. Following this, in the adjuvant setting, all the patients were treated at the same hospital using the standard of care regimens. None of them received immune checkpoint therapy. Two patients were enrolled in a Southwest Oncology Group (SWOG) clinical trial. These samples obtained from Baylor Scott and White Hospital were used for generating BSW-Discovery data and Nanostring Geo Mx spatial transcriptomics validation data.

In addition, for validation studies using IMC, five FFPE TNBC tissues, each from BA and WA, were obtained from Baylor Scott and White, and the data from these samples were termed BSW2 validation.

Also, 46 FFPE TNBC tissues (15 BA and 31 WA) from Roswell Park Comprehensive Cancer Center were used for IMC-based validation.

For 10X Spatial Transcriptomics, we examined five WA and four BA TNBC samples described above from the Baylor Scott and White (BSW2 validation) hospital. In addition, 10X spatial transcriptomics data for four WA and six BA samples from the Georgia cohort was obtained in collaboration with Dr. Aneja from the University of Alabama at Birmingham.

### Region of Interest (ROI) selection and segmentation
For each patient core (in the BSW Discovery cohort, 3 mm diameter) analyzed using imaging mass cytometry, 10X spatial transcriptomics or Nanostring Geo Mx Digital Spatial Profiling (DSP), ROIs were selected based on two parameters- tumor location (center vs periphery) and amount of immune infiltration observed in the H&E section (immune rich vs immune poor). This was performed by Dr. Asirvatham, breast pathologist and independently verified by Dr. Danika (resident pathologist, Supplementary Fig. 2). A total of 98 ROIs (each 600um-by-600um) were selected across the 8 TMAs (and 56 TNBC tissues), each belonging to one of four categories- Tumor Center Immune Rich (TCIR), Tumor Center Immune Poor (TCIP), Tumor Periphery Immune Rich (TPIR), and Tumor Periphery Immune Poor (TPIP). Supplementary Table 1 describes the number of ROIs belonging to each of these four groups across all the specimens analyzed. Importantly, the number of TCIR, TCIP, TPIP and TPIR analyzed were comparable across tumors from both BA and WA patients (Supplementary Table 1). Furthermore, H&E-based stratification of tumor-associated ROIs into immune-rich and immune-poor regions was confirmed by quantification of CD45 + immune cells (Supplementary Fig. 3). Notably, IMC and Nanostring GeoMx analysis were conducted on adjacent TMA sections containing identically marked ROIs.

The following guidelines were used to select the ROIs. TMA cores were at least 3 mm in diameter. Tumor periphery was defined as within 1 mm of the tumor perimeter when identifiable. The tumor center was defined as greater than 1 mm from the tumor perimeter. Stromal tumor infiltrating lymphocytes were determined using the International TILs Working Group 2014 guidelines[67] on H&E slides, following completion of available tutorials at tilsinbreastcancer.org. Less than 10% was considered immune poor, and greater than 50% was considered immune rich. A single pathologist (JRA) selected all ROIs at a single sitting. These ROI were verified by a second reviewer (PBD) with 100% agreement.

### Imaging mass cytometry (Mass CyTOF), BSW-Discovery data set
For the discovery data, a panel of 26 antibodies were selected based on the markers expressed in common immune cell populations- including T cells, macrophages, dendritic cells, and natural killer (NK) cells- as well as those expressed on tumor cells- including markers for hypoxia, angiogenesis, proliferation, epithelial mesenchymal transition (EMT),

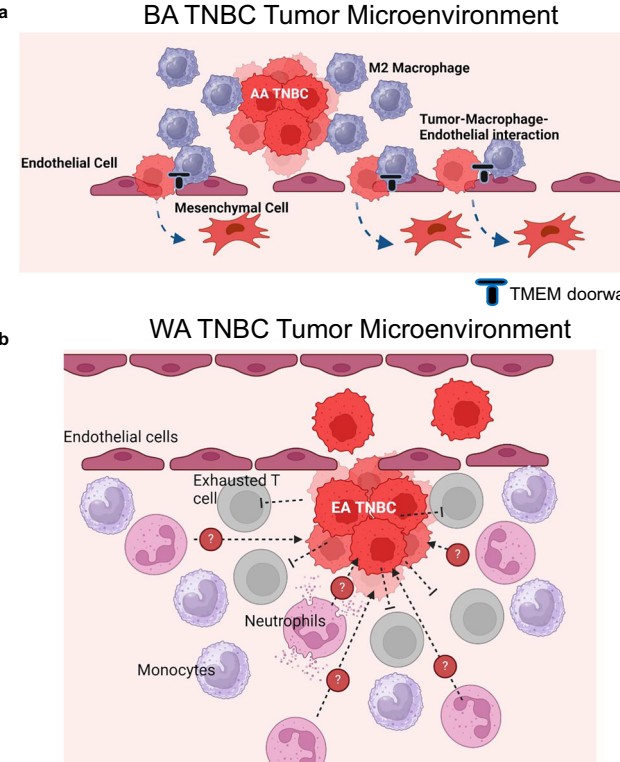

**a** BA TNBC Tumor Microenvironment

**b** WA TNBC Tumor Microenvironment

**Fig. 8 | Distinct Tumor Microenvironments (TME) in Black American (BA) and White American (WA) Triple Negative Breast Cancer (TNBC). a** BA Tumor Microenvironment (TME): The TME in BA TNBC is defined by a dynamic and highly integrated interaction between endothelial cells, macrophages, and tumor cells undergoing epithelial-mesenchymal transition (EMT). In this model, tumor cells express vimentin, a key marker of EMT, which enhances their motility and ability to invade surrounding tissues. A critical feature of the BA TME is the presence of a Tumor Microenvironment of Metastasis (TMEM), where the interplay between endothelial cells and macrophages creates specialized "doorways" that facilitate the intravasation of tumor cells into the bloodstream. These TMEM "doorways" are regions where the endothelial cells undergo modifications that increase permeability, enabling tumor cells to enter the vasculature more easily. The macrophages in this context are key players, secreting factors that promote endothelial cell destabilization and fostering the creation of these metastatic niches. The resulting tumor-promoting environment is immune-suppressive, limiting the ability of host immune cells to effectively target and eliminate the tumor. This BA-specific TME, combined with the TMEM mechanisms, supports tumor progression and the establishment of metastatic lesions. **b** WA Tumor Microenvironment (TME): In contrast, the WA TME is marked by a prominent immune exhaustion phenotype. Here, exhausted T-cells and neutrophils are prevalent, set against a backdrop of myeloid cells that drive immune suppression. This immune-suppressive environment is in part maintained by the interactions between exhausted T-cells and the cytokine milieu, which promotes immune dysfunction. While the WA TME does not exhibit the same TMEM "doorways" as seen in BA TNBC, the immune-exhaustive nature of this microenvironment still creates challenges for tumor-targeting immune responses. In this context, the accumulation of myeloid cells, particularly macrophages, perpetuates a cycle of immune suppression, preventing effective anti-tumor immunity and allowing the tumor to persist and potentially metastasize. Figure 8A created in BioRender. Sreekumar, A. (2025) https://BioRender.com/x2oyoly. Figure 8B created in BioRender. Sreekumar, A. (2025) https://BioRender.com/dka48j4.

and immunosuppression. In terms of the distribution of ROIs among patients in this BSW-Discovery cohort, a total of 98 ROIs were examined that included 47 ROIs from 26 BA patients and 51 ROIs from 31 WA patients. This included one BA patient with 3 ROIs, 19 BA patients with 2 ROIs and 6 BA patients with 1 ROI. For WA, we had 20 patients with 2 ROIs and 11 patients with 1 ROI.

Mass CyTOF antibody conjugation and staining were done in collaboration with the Aneja lab at Georgia State University. Lanthanide metal-labeled and Iridium intercalator antibodies were purchased from Fluidigm. Unconjugated antibodies were conjugated using the Max Par X8 labeling kits from Fluidigm. The concentrations of the conjugated antibodies were assessed using the NanoDrop system, and the final concentration was adjusted to 500 μg/ml. The conjugated antibodies were stored in an antibody stabilizer at 4 degrees Celsius. Descriptions of antibodies and isotope tags are described in Supplementary Table 4.

Staining of the tissues was performed as previously described[21,34,68]. Briefly, slides were de-paraffinized in xylene and rehydrated in alcohol. Antigen retrieval was performed with pre-heated Tris-EDTA buffer (pH 9) at 95 degrees Celsius in a de-cloaking chamber (Biocare Medical). The slides were cooled and blocked with 3% BSA (in PBS) for 1 h. Slides were then incubated with metal-tagged antibodies (1:50) overnight at 4 degrees Celsius. Counterstaining of the nuclei was performed with Iridium intercalator (1:200 dilution).

### Image acquisition by Hyperion and data analysis

Image acquisition was done in collaboration with the Flow Cytometry core at Baylor College of Medicine. Tissue analysis was performed using a Helios time of flight mass cytometer coupled to the Hyperion Imaging System (Fluidigm). Prior to acquisition, the imaging system was auto tuned using a 3-element tuning slide (Fluidigm) according to the manufacturer's instructions. ROIs for imaging were selected as described in Supplementary Fig. 2. Following the flushing of the ablation chamber with Helium, the tissue sections were ablated in a spot-by-spot fashion by a UV laser spot at 200 Hz frequency and 1μm resolution. The results were stored in Fluidigm's MCD format and exported as 16-bit OME TIFF format for downstream quantification.

### Imaging mass cytometry (Mass CyTOF), BSW-2 and Roswell Park Validation data sets

Here, 5 WA and 5 BA TNBC FFPE tumors (whole sections) were analyzed from the BSW-2 validation cohort, and 15 BA and 31 WA TNBC FFPE tumors (1 mm core size) were analyzed from the Roswell Park validation cohort (Summarized clinical data in Supplementary Table 2). Regions of Interest (ROIs) were selected by Drs. Asirvatham and Gutierrez, respectively, for the two cohorts. In the BSW-2 validation cohort, the ROI size was 700 μm by 700 μm. For the Roswell Park validation cohort, ROI size was 600 μm-by-600 μm, and each ROI covered the entire 1 mm core. The IMC analyses were done by the Immune Monitoring Core at the Houston Methodist Research Institute. Sample preparation commenced with the staining of tissues using pathologist-verified, metal-tagged antibodies, which were optimized for the CyTOF® imaging system[69]. A total of 27 and 32 metal-tagged antibodies respectively, were used to examine the BSW-2 and Roswell Park validation samples (refer Supplementary Tables 5 and 6) These antibodies allowed comprehensive analysis of immune, stromal, and tumor cell heterogeneity, in addition to various cell subsets and functional phenotypes within the tumor microenvironment (TME). All the antibodies were prepared according to the manufacturer's protocols provided by Standard Bio-Tools, measured for absorbance, and stored in Candor PBS Antibody Stabilization solution (Candor Bioscience) at 4 °C. FFPE tissue sections were subjected to baking, dewaxing in xylene, rehydration through graded alcohols, and heat-induced epitope retrieval in an EZ-Retriever System (BioGenex) at 95 °C using a Tris-Tween20 buffer at pH 9 for 20 min. After blocking with 3% BSA in TBS, the sections were incubated overnight with an antibody master mix, followed by washing, and staining for nuclear identification using Cell-ID Intercalator (Standard BioTools). Subsequently, the slides were washed, air-dried, and stored for ablation. The sections were then ablated using the Hyperion system (Standard BioTools) for data acquisition. Data acquisition followed the method described earlier for the Discovery data set.

## 10X Visium spatial transcriptomic profiling

Here, 9 WA and 10 BA TNBC FFPE tumors from two independent cohorts were analyzed (4 BA and 5 WA from BSW2 and 6 BA and 4 WA from the Georgia cohort, See Supplementary Fig. 14–17 for H&E images and ROI markings). The 10X spatial transcriptomics analyses were conducted by the Immune Monitoring Core at the Houston Methodist Research Institute for the BSW2 cohort, and the Advanced Genomics Core at University of Michigan for the Georgia cohort.

cDNA libraries were prepared following the guidelines outlined in the Visium CytAssist Spatial Gene Expression for FFPE User Guide. FFPE tissue sections of 5 μm thickness were mounted on Superfrost™ Plus Microscope Slides (Fisherbrand™) and subjected to H&E staining after deparaffinization. Following imaging, the cover slips were removed from the sections, and the sections were processed for hematoxylin de-staining and de-crosslinking. The glass slide bearing the tissue section underwent processing using the Visium CytAssist instrument (10x Genomics) to facilitate the transfer of analytes to a Visium CytAssist Spatial Gene Expression slide, which features a 0.42 cm$^2$ capture area with 4,992 spatial barcodes. Subsequent to this, the probe extension and library construction were carried out according to the standard Visium for FFPE workflow, following the manufacturer's protocol, but outside the instrument. The libraries were then sequenced using paired-end dual-indexing (28 cycles for Read 1, 10 cycles for i7 index, 10 cycles for i5 index, and 50 cycles for Read 2) on an Illumina NovaSeq X platform, achieving an average of 30,000 reads per spot. The resulting FASTQ files, together with the H&E images, were processed using Space Ranger version 2.1.0 (10x Genomics), using the GRCh38-2020-A reference genome.

## Gene expression analysis using GeoMX DSP system (NanoString)

The nanoString platform was used to profile the patient samples in the BSW-Discovery cohort, the same patient cohort used in the BSW-Discovery IMC data. Supplementary Figs. 1 and 2 describe the H&E images, selected ROIs and the immunofluorescence staining with PanCK and CD45 for each tissue. The latter was used as the morphology markers to segment the tissues into epithelial and immune cell compartments as described below. Supplementary Table 1 describes the various categories of ROIs selected by the breast pathologist, Dr. Asirvatham, for the Nanostring analysis.

We queried the transcriptome in these samples using the GeoMX Human Whole Transcriptome Atlas (WTA), which measures ~18,000 protein-coding genes. GeoMX DSP analysis was performed as previously described[70]. In brief, the 8 TMAs were stained with mRNA hybridization probes attached to UV-photo cleavable indexing oligonucleotides. The slides were stained with two morphology markers- Pan Cytokeratin (PanCK, to identify tumor cells), and CD45 (to identify immune cells), in addition to a nuclear stain (DAPI). Specific regions of interest (ROIs) were selected by the breast pathologist, Dr. Asirvatham (and confirmed by an independent pathologist, Dr. Binsol) under the guidance of the morphology markers. Each ROI was thus divided into three segments based on the positive/negative staining of the morphology markers- Tumor segment (PanCK + ve CD45 − ve), Immune segment (PanCK − ve CD45 + ve), and Stroma segment (PanCK − ve CD45 − ve). 293 segments (98 tumor segments, 98 immune segments, and 97 stromal segments) were generated in this fashion, from 57 tumors (26 BA and 31 WA tumors, respectively). Since the stromal segment was defined using negative selection, it contained a mixed population of cells, and hence was not considered for downstream analysis. The slides were exposed to UV light, thereby releasing the indexing oligonucleotides. The oligonucleotides from each ROI were then collected into specific wells of a microplate and counted using the nCounter system.

## Multiplex Immunofluorescence Analysis

This was used to validate the co-localization of endothelial cells and macrophages in BA TNBC. The samples used were the BSW-Discovery patient cohort, the same used to generate BSW-Discovery IMC data and Nanostring GeoMx DSP data. These tissues were distributed across 8 TMAs. Briefly, FFPE patient TMA slides were subjected to baking at 60 °C for two hours and washed with xylene to remove excess paraffin. Subsequently, the slides underwent rehydration by incubating in a series of ethanol solutions at various concentrations (100, 95, 70, 50, 30, and 0%). After rinsing with PBS, the slides were immersed in 1x Target Retrieval Solution, pH 9 (Dako, S2367) at 115 °C for 15 min in a pressure cooker for antigen retrieval. Following antigen retrieval, the tissue sections were permeabilized and blocked using 10% normal donkey serum in PBS-GT (2% fish gelatin, 0.1% Triton-X100 in PBS) for 1 h at room temperature. Primary antibodies (goat anti-CD31, R&D, AF3628; rabbit anti-CD163, Abcam, ab182422) were diluted to 1:100 in PBS-GT and applied to the tissue sections, which were then incubated in a humid chamber at 4 °C overnight. The following day, the slides were washed with PBS three times and incubated with secondary antibodies (Alexa Fluor 488-conjugated Donkey anti-Rabbit IgG, Jackson ImmunoResearch, 711-545-152; Alexa Fluor 555-conjugated Donkey anti-Goat IgG, Thermo Scientific, A-21432) diluted to 1:500 in PBS-GT for 2 h at room temperature.

After two washes with PBS, the slides were stained with Hoechst (20 μg/mL) in PBS for 5 min at room temperature, followed by two additional PBS washes. ProlongTM Gold antifade mountant was applied, and a cover slip was carefully placed. The slides were left to cure in the dark at room temperature overnight. Finally, images were acquired using a Zeiss LSM 780 confocal microscope, and the fluorescent images were processed using Zen software (Zeiss).

## Immunofluorescence quantification and co-localization

Multi-channel TIFF images were initially separated into individual color channels: Red – representing CD31, Green – CD163, and Blue – Hoechst. We conducted pixel-level co-localization between CD31 and CD163 channel images in ImageJ[71]. To begin with, the individual channel was smoothed using Gaussian Blur at the default setting. Then, signal intensities per channel were auto-thresholded to preserve the upper 5–10% of the image histogram. Signal intensities were segmented by using the Find Maxima function in ImageJ, and locations of segmented pixels of CD31 and CD163 intensities were recorded into the Region of Interest (ROI) Manager. Next, we computed co-localization frequencies between CD31 and CD163 segmented pixels by the following procedure. The number of pairs of pixels (where one from the CD31 pixel list and one from CD163) that are separated by a < 40-pixel Euclidean distance apart was quantified per image. This was repeated for all 103 immunofluorescent images of WA and BA TNBC tumors that were acquired.

## Imaging mass cytometry analysis

Our IMC data, stored in MCD files, were processed using the imctools to generate multichannel TIFF images for protein markers. For cell segmentation, we used tools such as HistoCAT[68], CellProfiler[72], and CellPose[73]. We started by applying a Gaussian Blur in ImageJ[74] to smooth the Ir191 and Ir193 DNA channel images, enhancing signal continuity within cell nuclei. Deep learning tool CellPose was applied to perform cell segmentation on DNA channels. For images with unsatisfactory segmentation, ilastik[75] was used for semi-supervised pixel classification; a few cells were manually segmented to train the classifier for segmenting the entire image. HistoCAT uses cell masks to compile a cell-by-protein matrix, summarizing pixel intensities for each cell and protein marker. Cell position was determined by the centroid position of cell masks obtained from DNA channel segmentation.

## Clustering and differential protein analysis

We applied log-transformation and z-scoring, initially across all cells and then across all protein markers. Next, we used K-means clustering

with a high number of random starts (nstart = 100,000) for reliable centroid generation. This was used because K-means promoted greater tolerance for noise and variation in data processing, was less sensitive and more robust to outlier expression, and avoided the generation of ROI (region of interest)-specific clusters. We determined 20 K-means clustering showing robust cluster-specific expression. We next determined differentially expressed proteins across clusters by performing all one-vs-one comparisons. Statistical significance was reached if $P <= 0.05$. We deemed a protein to be differentially expressed for a cluster $C_k$ if the protein is significant in 17 out of 19 one-vs-one comparisons between $C_k$ and $C_n$ where $n = 1 \ldots N$ excluding $k$. After that, each cluster was named by the list of differential markers expressed in the cluster.

## Spatial interaction analysis in IMC

We constructed a Delauney spatial graph for single cells in each ROI and used it with cluster labels for spatial proximity analysis in the Giotto pipeline[35]. Giotto's Cell Proximity Enrichment tool, coupled with 1000 simulations, identified cell-type interaction enrichment or depletion compared to random simulations with shuffled labels. This analysis generated a z-score for each cell-type pair, indicating enrichment (positive z-score) or depletion (negative z-score) of interactions per ROI. Giotto is a validated pipeline for inferring spatial interactions from Multiplexed Ion Beam Imaging (MIBI), IMC and compatible data. We have previously illustrated an example of analyzing an MIBI TNBC dataset[22] in the Giotto paper[35]. Next, we conducted group-wise comparisons to extract racial group-associated cell-cell interactions (CCIs). For this analysis, race groups were self-reported Black American (BA) or White American (WA) TNBC patients. We used a mixed linear statistical model to find the racial group associated CCIs. The linear mixed model accounts for variations derived from patients while allowing each patient to have repeated observations in the form of multiple ROIs per patient[76]. This approach yields more accurate $P$-values than standard tests that falsely assume independence between ROIs. We used the lme4 R package for this analysis[76]. In this linear model, we write the design function as "$score + race + (1|patient)$", where $score$ is the interaction score of a particular cell-type pair in ROI, $race$ is either BA or WA, and $patient$ is the patient identifier of ROI. Multiple ROIs are associated with each patient (i.e., making $patient$ the random effect), whereas $race$ is the main (or fixed) effect. The alternative null model has the design function "$score \sim 1 + (1|patient)$" which excludes the race-associated effect. ANOVA testing was performed between these two models to select only the interaction pairs whose scores are most affected by race.

## Cell Community detection in IMC

Racially distinct cell-cell interactions that pass the FDR cut off 0.20, with a positive BA-coefficient (for BA-specific CCI) or positive WA-coefficient (for WA-specific CCI) in the linear model, were selected to form cell communities. For the selected race-specific CCIs, $X$, consisting of interactions $a-b$, where $a, b$ are cell type clusters, we form a graph $G$ where nodes are the clusters, and the edges connect two clusters if there is a CCI between them. We next obtain connected components of $G$, which will result in an initial set of cell communities. Each community will undergo pruning steps: If a community consists of just one node with self-interaction, it is removed. If a community consists of weak "bridging" edges between sub-communities, these bridging edges are removed, and the resultant sub-communities will become communities. Such bridging edges are usually apparent from the topology of the community graph, since there is usually one bridging edge connecting two sub-communities (if sub-community structure exists within the community).

## Community matching between IMC datasets

Cell community matching is possible between the 3 IMC datasets utilized in this study, since all datasets use the same or similar protein antibody panel with >90% of antibodies shared across datasets. Thus, to match cell communities that are found independently from each IMC dataset, each community is initially reduced to a set of member cell clusters and further reduced to these clusters' differentially expressed protein markers. We next performed matching at the protein marker levels. The two communities (one in each IMC dataset) with the highest number of protein marker matches are declared a matching community. The top-matched communities are termed BA-Community1 and WA-Community1, respectively, for each race. In the special case where a community in dataset 1 looks to be a union of two communities in dataset 2 (which would produce an equally likely match to each of two communities), we will go to dataset 3 to decide whether or not to split the community in dataset 1 or join the two smaller communities in dataset 2 in matching with dataset 1. BA- and WA-Community1 represent the top-matching community across the 3 IMC datasets. Weaker scoring matches are termed Community2, 3, etc. Dataset-specific communities are ranked last.

## Pre-processing of 10X visium spatial transcriptomics data

We have used Space Ranger to align the reads to human hg38. Following this, we loaded the resulting HDF5 file into Giotto for preliminary analyses. This includes dimensionality reduction, UMAP (Uniform Manifold Approximation and Projection), and KNN-based Leiden clustering. We preprocessed each spatial transcriptomics sample in Bassiouni et al.[30] and in our own TNBC cohorts (used for validation) to generate normalized gene expression matrices per ST sample with Giotto[35].

## Targeted analysis of spatial transcriptomic datasets

**Query co-localization in ST dataset.** Each community was converted from protein names to gene names. The list of gene names in the community forms a race-specific query, such as BA-community 1. For a given query, we computed a query co-localization score (QCS) per spot $x$ per ST sample $A$ by summing the scaled and log-normalized expression of query genes in the spot:

$$QCS_{x,A} = \frac{1}{|Q|} \sum_{q \in Q} Expr_{x,A}(q) \tag{1}$$

$QCS_{x,A}$ was deemed significant if it exceeded ($P < 0.05$) the summed score of the randomly shuffled case, whereby the expression of each query gene was randomly distributed among all the in-tissue Visium spots in the sample and thus destroys the dependence among query genes. A per-sample $QCS$ score is next quantified as $QCS_A = |QCS_{x,A} > QCS_{shuffled,A}|$ which is equal to the total number of Visium spots with significant $QCS$ scores. This $QCS_A$ was compared between the BA and WA groups.

## Spatial clustering

In addition to $QCS$, we also quantified a Spatial Clustering score and conducted cross-query correlation analysis. In the former, to compute Spatial Clustering, we adopted the silhouette coefficient metric previously described in Zhu et al.[77] that measures the spatial coherence of gene expression pattern. Here, the pattern was represented by spots showing significant $QCS$ (given the label 1) against the background of remaining spots (given the label 0). Spatial Clustering ($\delta$), computed by silhouette coefficients, was next assessed individually for each query (i.e., BA-Community-1):

$$\delta = 1/|L_1| \sum_{s_i \in L_1} (m_i - n_i) / \max(m_i, n_i) \tag{2}$$

This silhouette coefficient[78] assesses the spatial distance associated with two sets, $L_1$ (spots given label 1) and $L_0$ (spots given label 0). For a given spot $s_i$ in set $L_1$, $m_i$ is defined as the average distance between $s_i$ and any spot in $L_0$, and $n_i$ is defined as the average distance

between $s_i$ and any spot in $L_1$. For distance, we used rank-normalized, exponentially transformed distance, which prioritizes local physical distance between two spots. The distance between spots $s_i$ and $s_j$ is defined as $r(s_i, s_j) = 1 - q^{rank_d(s_i, s_j) - 1}$ where $rank_d(s_i, s_j)$ is the mutual rank[79] of $s_i$ and $s_j$ in vectors of Euclidean distances $\{Euc(s_i, *)\}$ and $\{Euc(s_j, *)\}$. $q$ is a rank-weighting constant, set at 0.95. Spatial clustering was then compared between the BA and WA patient groups.

## Derivation of extended signature genes of BA- and WA-associated communities

To derive extended signature genes (ESGs) of BA- and WA- race-associated communities, for each community query (say BA-Community-1), we stratified the Visium spots per ST sample into interact-high and interact-low groups based on the $QCS$ score of that community query per spot. The community-high group consists of Visium spots in the top 10% of spots with the highest $QCS$. The community-low group consists of spots in the bottom 10% with the lowest $QCS$. These high- and low-groups were defined per ST sample. Per-sample differential gene expression analysis was next performed between the community-high and community-low spots within the ST sample, forming Extended Signature Genes of the query. Recurrent BA-associated ESGs were next derived by requiring ESGs to be present in at least 50% of BA patient samples, and the percentage of BA patients with ESGs as a signature must be higher than the percentage in WA patients.

## Cell-type specific expression of extended signature genes of BA- and WA-associated niches

We integrated single-cell RNA sequencing data of breast cancer atlas[80] with spatial transcriptomics to derive cell-type specific gene expression profiles in BA- and WA- interaction-targeted niches. The cell-type specific expression of gene $g$ in cell type $c$, in a niche $N$ that is defined by WA or BA-Community, shortened as $Expr_{g,c,N}$, was derived from:

$$Expr_{g,c,N} = AvgExpr_{g,c} \times V_{g,N} \qquad (3)$$

where $AvgExpr_{g,c}$ is the average expression of $g$ in cell type $c$ in the scRNAseq data, and $V_{g,N}$ is the ST average-subtracted expression of $g$ in spot $s$ in niche $N$ in 10X Visium sample. Niche is made up of spots in the community-high group defined earlier. Briefly, $V_{g,N}$ is defined as:

$$V_{g,N} = \sum_{s \in N} VExpr_{g,s} / |N| - \sum_{s \in S} VExpr_{g,s} / |S| \qquad (4)$$

Here, $VExpr_{g,s}$ is the Visium expression of $g$ in spot $s$. $V_{g,N}$ is the result of subtracting the background expression (summed over all spots $S$ in the ST sample) from the niche expression (summed over only spots in niche $N$). For cell types, we iterated over all Cancer Associated Fibroblasts (CAF), endothelial, Peri-Vascular Like (PVL) cell types, as well as all subsets of immune cells in Wu et al.[80], and Wigerblad et al.[81], and Alvarez-Breckenridge et al.[82], which provided the neutrophil subsets.

## Ligand-receptor pair analysis

We download all human ligand-receptor pairs from CellPhoneDB database[83]. We next computed enrichment of ligand-receptor pairs within ESGs of BA- and WA-associated cell niches by identifying pairs where both the ligand and receptor genes were present within the ESGs. This information was mapped to different niches, BA-Community-1 and WA-Community-1. Ligand-receptor pair results were next summarized into a cell-type interaction network as shown in Fig. 6a by the following procedure. The cell-interaction score $CI$ between two cell types $c_1$ and $c_2$ is given by:

$$CI(c_1, c_2) = \sum_{LR} \frac{I(LR, c_1, c_2)}{N(LR)} \qquad (5)$$

Where $LR$ is a ligand-receptor pair – we iterated over all ligand-receptors that are enriched among the ESGs. $I(LR, c_1, c_2)$ is an indicator function that is 1 if $LR$ is expressed in $c_1$ and $c_2$, and 0 otherwise. $N(LR)$ is the total number of cell type pairs in which ligand-receptor is expressed, defined as:

$$N(LR) = \sum_{c_1} \sum_{c_2} I(LR, c_1, c_2) \qquad (6)$$

## Overlap analysis with existing TNBC subtype gene signatures

Inflamed, Excluded, and Ignored signatures of TNBC were downloaded from Hammerl et al.[53] supplementary material. To compare these existing phenotype signatures with our WA-niche associated ESG signatures, we computed the number of overlapping genes and its statistical significance using the hypergeometric distribution test.

## Gene-set enrichment analysis of niche-genes

We used the g:Profiler[84] web server to compute gene-set enrichment statistics against the gene-set database[85] compiled by the Bader lab. This database has compiled all current GO Biological Process, and all pathway gene sets and continuously updated. We downloaded Human_GOBP_AllPathways_no_GO_iea.gmt and uploaded it directly to g:Profiler for enrichment analysis of BA- and WA-associated ESGs. Default enrichment settings were used for the analysis.

## NanoString GeoMx DSP validation analysis

We interrogated the CD45+ immune and PanCK epithelial compartment gene expression matrices to compare the expression level of race-associated niche genes in TNBC BA and WA patients (see Supplementary Figs. 1 and 2). For each patient and each ROI, a summary score was computed for each niche-associated ESG gene set based on summing log-normalized gene expression for all genes in each niche-gene set. The score was compared between BA and WA samples, and $t$ test statistics (testing WA > BA or BA > WA depending on the niche) were computed.

## 10X Visium spatial transcriptomics validation analysis

We validated the niche-specific ESGs by asking whether these ESGs are higher in BA patients or higher in WA patients in two independent validation cohorts (19 TNBC, 10 BA and 9 WA, see Supplementary Figs. 14–17). Specifically, for each community query, BA-Community-1 and WA-Community-1, we extracted community-high and -low spots per ST sample in our validation cohort. Next, for the corresponding niche specific ESGs, for each gene $g$ we computed

$$\log_2 \left( \frac{avgExpr_{g, high}}{avgExpr_{g, low}} \right) \qquad (7)$$

using the high and low spots that were defined and compared this log-fold change in BA samples and WA samples in validation cohort.

## Kaplan Meier (KM) survival analysis

Survival analysis was performed using the survival R package and visualized using the ggplot2 package autoplot function. In the BSW-Discovery and Roswell Park cohorts, survival analysis was done independently per cohort based on IMC-derived community profiles observed in patient ROIs. Specifically for each patient, we derive a per-community-score as follows: we compute the sum of interaction scores of the constituent cell-cell interactions of the community, deriving a community score. Then we sum the community scores of all BA-identified communities or WA-identified communities, deriving a community-all score per BA-communities or WA-communities. Patients are next stratified based on whether the community-all score of each patient is higher or lower than the average across all patients.

Kaplan Meier curves were next created using the survival library in R. We further examined the association between BA- and WA-associated communities with overall survival in a univariate and multivariate setting using Cox Proportional Hazard Analysis (Supplementary Table 2). In the latter, we controlled for age, race and stage.

## Cox proportional hazard analysis

We are interested in associating the BA-community and WA-community scores with patients' survival in the BSW-Discovery and Roswell Park Validation cohorts. Each cohort has documented patient attributes such as age, stage, race, BMI, and mutation status, in addition to 10-year survival status. We built both univariate models to examine the association of BA-/WA-Community scores to survival, and multivariate models to examine these associations while accounting for covariates such as age, stage, and race. In the univariate setting, the association was performed by the coxph function in R by setting "$Surv(time, status) \sim BAscore$" where BAscore is the BA-Community score for the individual patient. In the multivariate setting, the association was made by setting "$Surv(time, status) \sim BAscore + bmi + age + stage + race$". This examines the association of BAscore while accounting for other covariates. Both the aforementioned simple model and the race-stratified Cox model were attempted (e.g., "$Surv(time, status) \sim BAscore + bmi + age + stage + strata(race)$"), and similar coefficients and $P$ values were obtained.

## Visualization

Spatial gene expression profiles were plotted using the scatter function in the Python matplotlib library. Correlation maps were plotted with the seaborn library, the heatmap function, and the plasma colormap. Gene expression heatmaps were plotted with the seaborn library clustermap function with colormap set to Spectral. Protein co-localization figures (IMC) were generated for defined cell clusters using the scatter function. Lastly, we used the lineplot function (seaborn package) to compare gene expression between WA and BA groups.

## Reporting summary

Further information on research design is available in the Nature Portfolio Reporting Summary linked to this article.

## Data availability

All data supporting the findings of this study are available within the paper and its Supplementary Information. The 10X VisiumSpatial Transcriptomic data of TNBC patients, including the minimum dataset from the Georgia cohort, has been deposited in Zenodo under the following URL: https://doi.org/10.5281/zenodo.12797059. The Nanostring GeoMx data has been deposited into Zenodo under the following URL: https://doi.org/10.5281/zenodo.12752405. The Imaging Mass Cytometry (IMC) data has been deposited into Zenodo under the following URL: https://doi.org/10.5281/zenodo.15115492. Source data are provided in this paper.

## Code availability

The codes used to analyze the TNBC disparity data in this study were developed using R, Python and Shell. The codes used to analyze the data in this study are deposited in GitHub under the URL link https://qianzhulab.github.io/suppl/TNBC.scripts/table_of_content.html. DOI has been created from Github: https://doi.org/10.5281/zenodo.15353111.

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

## Acknowledgements

This study was partly funded by RO1CA227904 (ASK and XZHF) and 1 P20 CA284971-01 (ASK), and a Cancer Prevention Research Institute of Texas (CPRIT) grant RR220035 (QZ). Dr. Zhu is a CPRIT scholar in Cancer Research. Dr. Zhang is a McNair scholar. Dr. Sreekumar is supported by the Charles C Bell Jr-endowed professorship. ECC is also supported by the following grants from the DoD: W81XWH-21-1-0106, W81XWH-21-1-0634, and HT94252410103, and P50CA186784 from NIH. The authors acknowledge Dr. Liang Zhang and Kathy Ton from the NanoString TAP team for performing the GeoMX Human Whole Transcriptome Atlas assay and assisting with the initial data analysis. IMC data collection was supported by the Cytometry and Cell Sorting Core at Baylor College of Medicine with funding from the CPRIT Core Facility Support Award (CPRIT-RP180672), the NIH (CA125123 and RR024574) and the assistance of Joel M. Sederstrom. Confocal microscopy was supported by the Integrated Microscopy Core at BCM with funding from NIH (DK56338 and CA125123), CPRIT (RP150578), the Dan L. Duncan Comprehensive Cancer Center, and the John S. Dunn Gulf Coast Consortium for Chemical Genomics. The research was also supported by NCI grant P30CA016056 (Cancer Center Support Grant Development Funds) to the Biorepository and Laboratory Services Shared Resource at the Roswell Park Comprehensive Cancer Center, and P30CA013148 to the O'Neal Comprehensive Cancer Center at the University of Alabama at Birmingham. This work was also supported by CPRIT RP210027 - Baylor College of Medicine Comprehensive Cancer Training Program to Megha Chatterjee. The authors acknowledge Callee Arnold, Histo-technologist at BSWH, for technical expertise.

## Author contributions

A.B., M.C., J.K., L.W., M.B., U.R., Y.X., and E.H.S.: collected experimental data. B.P., S.W., and J.Z. pre-processed IMC, 10X ST and Nanostrong GeoMx DSP data. D.J., A.R., and J.R.A. created the TMAs. JRA selected the patient samples. J.R.A. and P.D.B. marked the ROIs for IMC, 10X ST and Nanostring GeoMx DSP analysis for the BSW discovery and BSW2 validation. C.G. marked the ROIs for IMC for the Roswell Park validation. N.M. and N.P. compiled the clinical data for the TNBC samples from the BSW cohort. J.R.A., S.-h.C., X.L., A.R.O., C.M., G.M.D., and C.A. provided reagents and resources. RA provided processed 10X data for the Georgia validation cohort. R.A., Y.L., E.C., E.B., and X.H.F.Z. Critically reviewed the data and provided scientific inputs. Q.Z. developed and performed all the original integrative analysis, conceptualized the study. A.S.K. conceptualized, directed, and funded the study. All authors discussed the results and commented on the manuscript.

## Competing interests

Jaya Ruth Asirvatham served as an advisor for Roche. Xiaoxian Li has served as an advisor for AstraZeneca, Roche, Eli Lilly, and Onviv, and Champions Oncology has funded research in Xiaoxian Li's lab. Dr. Sreekumar reports grants from the Agilent Foundation, non-financial support from Sri Sathya Sai Institute for Higher Learning, India, and personal fees from Karkinos Health Care Pvt. Ltd., India, outside the submitted work. The remaining authors declare no competing interests.

## Additional information

Qian Zhu [1,2,3,15] ✉, Akhila Balasubramanian[4,15], Jaya Ruth Asirvatham[5,15], Megha Chatterjee [4], Badrajee Piyarathna[4], Jaspreet Kaur[6], Nada Mohamed[5], Ling Wu [1], Stacy Wang[1,2], Niloufar Pourfarrokh[5], Paula Danika Binsol[5], Mahak Bhargava [7], Uttam Rasaily[3], Yitian Xu[8], Junjun Zheng[8], Deborah Jebakumar[5], Arundhati Rao[5], Carolina Gutierrez [1,3], Angela R. Omilian[9,10], Carl Morrison[9], Gokul M. Das [11], Christine Ambrosone [10], Erin H. Seeley[12], Shu-hsia Chen[8], Yi Li [1,3,4], Eric Chang [1,4], Xiaoxian Li [13], Elizabeth Baker[14], Ritu Aneja[7], Xiang H.-F. Zhang [1,3,4] & Arun Sreekumar [3,4] ✉

¹Lester and Sue Smith Breast Center, Baylor College of Medicine, Houston, Texas, USA. ²Department of Human Molecular Genetics, Baylor College of Medicine, Houston, Texas, USA. ³Dan L Duncan Comprehensive Cancer Center, Baylor College of Medicine, Houston, Texas, USA. ⁴Department of Molecular Cellular Biology, Baylor College of Medicine, Houston, Texas, USA. ⁵Department of Pathology, Baylor Scott and White Health, Temple, Texas, USA. ⁶Department of Biology, Georgia State University, Atlanta, Georgia, USA. ⁷Department of Nutrition Sciences, School of Health Professions, The University of Alabama at Birmingham, Birmingham, Alabama, USA. ⁸Immune Monitoring Core, Houston Methodist Research Institute, Houston, Texas, USA. ⁹Department of Pathology and Laboratory Medicine, Roswell Park Comprehensive Cancer Center, Buffalo, NY, USA. ¹⁰Department of Cancer Prevention and Control, Roswell Park Comprehensive Cancer Center, Buffalo, NY, USA. ¹¹Department of Pharmacology and Therapeutics, Roswell Park Comprehensive Cancer Center, Buffalo, NY, USA. ¹²Department of Chemistry, College of Natural Sciences, The University of Texas at Austin, Austin, Texas, USA. ¹³Department of Pathology and Laboratory Medicine, Emory University, Atlanta, Georgia, USA. ¹⁴Division of Molecular and Translational Biomedicine, Department of Anesthesiology and Perioperative Medicine and Division of Pulmonary, Allergy, and Critical Care Medicine, The University of Alabama at Birmingham, Birmingham, Alabama, USA. ¹⁵These authors contributed equally: Qian Zhu, Akhila Balasubramanian, Jaya Ruth Asirvatham. ✉e-mail: qian.zhu@bcm.edu; arun.sreekumar@bcm.edu

