## [Peer Review file · Nature Communications]

Integrative spatial omics reveals distinct tumor-promoting multicellular niches and immunosuppressive mechanisms in Black American and White American patients with TNBC

Corresponding Author: Dr Arun Sreekumar

Version 0:

Reviewer comments:

Reviewer #1

(Remarks to the Author)

The paper presents a study on triple-negative breast cancer (TNBC) that reveals distinct spatial cell-cell interactions between tumors from self-reported Black American and White American women. Despite using a racially balanced and clinically matched cohort, the study found no significant differences in overall survival rates or mutational statuses of key genes between the two groups. However, the spatial organization and interactions within the tumor microenvironment (TME) were racially distinct, with BA tumors showing more endothelial-macrophage-mesenchymal interactions and WA tumors displaying exhausted T-cell and neutrophil interactions. While the study is of importance and the findings are of interest there are significant concerns with the interpretation of the data by the authors in addition to the clinical translation and broader impact being limited.

Overall:

Introduction:

This section is a little long and somewhat repetitive. Also, citations are not up-to-date. Suggest refining this paragraph and working on more robust citations. Specific comments as below:

- The utilized references 9 and 10 do not support this statement by the authors "Importantly, cancer-related disparities in mortality between BA and WA women was higher for TNBC than other subtypes, even after adjusting for age, stage, treatment, socio-economic status, poverty index, and treatment delay". The Caggiano article shows that adjusting for clinical factors including subtype makes mortality disappear, and the other study shows worse survival amongst Black patients overall not specifically for those with TNBC. Suggest using more robust references here, including published SEER data showing disparities in TNBC.
- The references used in the introduction to cite literature on this topic are not current or well supported; suggest doing a more extensive literature review and citing more recent articles to support the background presented in the introduction. For example, the authors report "Many studies have examined the genetic component to understand the biological underpinnings of racial disparity", yet cite only one study in that regard. Other relevant studies include: Huo D et al. JAMA Oncol. 2017;3:1654–62. Newman LA et al. Ann Surg. 2019;270:484-492. Abdou, Y., et al. Breast Cancer Res 22, 62 (2020)
- Other relevant studies on variations in TME between Blacks and Whites include: Abdou, Y., et al. Breast Cancer Res 22, 62 (2020). Marczyk M. et al. npj Breast Cancer 8, 88 (2022).
- Suggest providing reference for this statement "the experience of racism and being a member of a racialized group differentiates the lived experience of BA and WA women in multiple ways, all of which together result in biological differences."

Major Concerns:

1. This manuscript used regions of interest evaluated from the tumor samples using imaging mass cytometry and selected by a breast pathologist. This appears to be done based on the location of the ROI and the terms "immune rich" vs. "immune poor." Were those terms evaluated using antibody-based staining of immune cell markers in the tumor microenvironment (TME) or just by an H&E evaluation of cells that appear to be immune cells? Did the authors have a second pathologist evaluate the sections for standardization? This is critical as these findings were not completely recapitulated using 10X analysis of whole tumors.

2. The initial evaluation of cell clusters in figure 1B is somewhat difficult to assess. For instance, cluster 12 is composed of cells expressing PD-L1, CD16 and CD163, which are found on a number of different cell types including macrophages, monocytes, NK cells and some dendritic cells. Thus, what is this cluster and is it composed of all of these cells, or a subset of them. How do the authors assess cellular interactions for these clusters when the cells that are potentially present are this broad? Similarly, although B cells have been clearly found in TNBC samples (Conte et al. EBioMedicine 2024), B cells are not clustered from the TMAs. Why is that? It would be helpful for the authors to take one of the TMAs and stain this with standard antibodies (CD4, CD8, CD19, CD14, CD11c) and compare the cells identified with the cell clusters identified by mass cytometry.

3. This reviewer is quite confused on how the Giotto spatial network function and Delaunay network can be used for evaluating differences between patients based on race given the differences in cellularity across the samples. Giotto is typically used to infer cellular relationships by using a triangulated grid to infer cellular interactions, so it is unclear how the authors are comparing specimens with significant immune infiltration versus those without this. Presumably the latter will have far fewer cellular networks but could still have productive interactions between cells. Is this being evaluated? How do the authors differentiate racial differences in CCI between racial differences in cellular heterogeneity or abundance? This reviewer strongly believes that the best way to do this analysis would be to evaluate amongst the FoxP3+ population how many of these interact with DCs (for example) independent of the total number of FoxP3+ cell interactions. If that were done, are there still different interactions between the patient groups? It's not clear to this reviewer that the Giotto spatial evaluation can be used to compare interactions across tissues but should be used to infer potential interactions within a tissue. This was done in Figure 2H but not done for any of the other purported cellular interactions. Finally, it's not clear to this reviewer why the authors believe that the CD163+ cells are macrophages and not blood monocytes interacting with endothelial cells as part of their migratory activity.

4. The robustness of the spatial interactions in the current data set is unclear to this reviewer. Thus, is there significantly more prognostic information obtained from the regional assessment of cell expression compared to overall single cell expression from the sample. The authors insinuate that significant cell-cell interactions are driving the poor outcomes in different racial groups. However, it is not clear to this reviewer whether this is true or that the presence of immunosuppressive macrophages or regulatory T cells are driving the outcome independent of the cells they interact with. Can the authors confirm that there is significantly more prognostic information obtained from performing spatial assessment compared to only single cell assessment? If this were true, one would infer that the authors could demonstrate downstream changes in cells that are interacting with other cells from the spatial transcriptomic data. Was this true?

5. For the confirmatory data shown in figure 7, the authors need to show all of the extended signature gene groups. Did the additional platforms verify the BA-Q1 or Q2 ESG's? Similarly, did they verify the WA-Q1, Q4 ESG's? If they did not, what is the author's interpretation of those findings? Does that question the validity of these ESG's as being identified differently in the different racial groups with triple negative breast cancer? And if the authors argue that the differences are due to ROI vs. whole tumor assessment, isn't the whole tumor evaluation more robust? This suggests to this reviewer that there is significant bias in the assessment of the ROIs using imaging mass cytometry

6. There was almost a significant difference in the number of low-grade tumors in the two racial groups. The authors need to perform their analysis removing the low-grade tumors and only analyzing the grade three tumors. Were the findings still significant once these lower grade tumors were removed? Similarly the metastatic lesions should be removed from the analysis given differences in the immune response in metastatic lesions. Why would the authors choose not to exclude the case with HER2 positive disease amongst Black patients, since this study focuses on TNBC and given significant differences in clinical, immune and molecular characteristics between TNBC and HER2 positive, the exclusion of this case may be warranted? Suggest grouping stage into 1,2,3,4 rather than dividing into subgroups within each stage. This is for simplicity, and also, clinically the subgroups do not add to the significance of the analysis

7. The author should analyze the triple negative breast cancer groupings with the more granular assessment of intrinsic genes that separate out specific tumors as described in the discussion and published in the literature. It would then be helpful to try to ascertain whether there were specific features of the racially profiled tumors that were associated with basal-like immunomodulatory or the mesenchymal-like subtype. Finally, it would be quite helpful for the robustness of this manuscript if the authors were able to confirm cellular interactions leading to changes in gene transcription. Can they for instance demonstrate differences in macrophage activation based on their interaction with CD 31 expressing endothelial cells?

8. Table 1 describes that majority of cases (22 vs 29) did not receive chemotherapy. Given majority of cases are TNBC, this reviewer finds it odd that patients did not receive chemotherapy as part of their treatment paradigm. Did the authors mean to say that the tissue was analyzed prior to receipt of chemotherapy? If so, suggest clarification as chemotherapy would drastically affect outcomes in this setting. Finally, for the survival analysis, did any of the patients receive immune checkpoint inhibitor therapy, which is standard for neoadjuvant therapy for patients with TNBC and would confound the survival interpretation.

9. The authors highlight BMI differences between Blacks and Whites; although this did not translate into survival differences in Table 2, it would still be interesting to look at the interaction between BMI and the genomic findings highlighted. Also, this should be adjusted for in survival analysis, given BMI is a known contributing factor to disparities in outcomes as previously published

Minor:

1. The authors indicate that p53 and PIK3CA status did not differ between the cohorts and indicate that this is in Table 1 but

this information is found in Table 2. It would also be important to show these data as VAF fraction and mutation incidence rather than HR as indicated in the table.

2. This reviewer was not able to identify the statistical tests used for comparisons in figures 2F, G and H. This information should be provided in the legend to the figure.

3. It is assumed by this reviewer that the columns in figures 4C and E represent the different subgrouping of interactions shown in figure 4B. Is this correct?

4. It was somewhat surprising that the increased cell to cell interactions in the cohort of white patients was associated with an inferior prognosis given the presence of adaptive immune cells in those samples. This has also been more associated with black patients with TNBC (Yao S et al. J Natl Cancer Inst. 2021 Aug 2;113(8):1036-1043. doi: 10.1093/jnci/djaa215. PMID: 33395700; PMCID: PMC8328978.). A more elaborate discussion regarding this would be helpful Did any of these patients receive immune checkpoint inhibitor therapy? Was this treatment associated with a different outcome?

5. Typo noted in results section “ However, WA patients tend to have a higher risk for poor OS when stratified by WA-associated interactions (see WA-CCI * WA race in Table 2, HR 1.46)”. HR in table is 1.56. please specify which is correct HR.

6. Without additional information, it is highly speculative and discussion section for the authors to argue that racial differences in the immune landscape of triple negative tumors should be investigated using genetically engineered mice. It is entirely unclear to this reviewer how these mice can inform on “racial differences” in the immune landscape without additional information regarding what processes are driving this finding.

(Remarks on code availability)

Reviewer #2

(Remarks to the Author)

(Remarks on code availability)

Reviewer #3

(Remarks to the Author)

The manuscript titled “Integrative spatial omics reveals distinct tumor-promoting multicellular niches and immunosuppressive mechanisms in Black American and White American patients with TNBC” is a mainly well written and thorough paper on a highly relevant topic that has received too little attention previously. Racial disparities in the immune microenvironment of triple negative breast cancer can explain some of the differences in outcome between black and white women. In this manuscript the authors address this topic utilizing several methods exploring the immune tumor microenvironment, including imaging mass cytometry, immunofluorescence and different methods of spatial transcriptomics.

This manuscript is comprehensive and includes a lot of analyses and results, to a point where it is overwhelming to read, and it is difficult for a reader to keep track of the main message. I would highly recommend thinking about what MUST be included to keep the most important messages from drowning and try to strip the manuscript of unnecessary analyses. Figures 1-4 should be included, however, from there, I would recommend making only 1-2 figures with the most important results and also try to limit the number of supplementary figures. This said, the manuscript can absolutely become publishable if rewritten and shortened.

The introduction is well written and the material and methods sections include sufficient detail. The figures are not always informative, especially figure 6 and 7.

In addition I have some minor comments:

- It needs to be explained whether IMC and GeoMx is run on different slides. If they are run on different slides, are the ROIs for IMC and GeoMx the same/in the same area or different areas?
- I would recommend including an image of an GeoMX ROI showing segmentation into the AOIs.
- Suppl table 1: MSC (abbreviation should be explained). Messenchymal – one “s” to many
- Figure 1D – color legend needs to be added
- Supplementary figure 3A: the colors used in the dendrogram needs to be explained
- o It could be an idea to cluster the data using the relative abundance of immune cells only. In that way, the different abundance of tumor cells would not influence the clustering and it might uncover differences between BA and WA?
- Supplementary figure 3B: where is the 1 TPIP ROI described in Table 1? Coretype needs to be mentioned in the figure text.
- Figure 2F – the Zoom for the selected area should preferably be larger
- Figure 3E: Need to annotate that the 5 columns represent the 5 different queries
- Figure 5 could be a supplementary figure

- Figure 6 and figure 7 need to be better explained and justified.

(Remarks on code availability)

The code is accessible, however I'm not sure that it will be enough to reproduce by others.

Reviewer #4

(Remarks to the Author)

In the manuscript entitled "Integrative spatial omics reveals distinct tumor-promoting multicellular niches and immunosuppressive mechanisms in Black American and White American patients with TNBC", the authors seek to identify the biological mechanisms that potentially explain why Black American women with triple negative breast cancer (TNBC) have worse prognosis compared to White American women. They utilize a spatial omics approach that integrates spatial single-cell level immunohistochemistry and spatial transcriptomics data to identify race-specific niches that potentially correlate with overall patient survival. By integrating these different technologies, this research provides a more comprehensive approach that could lead to new insight in understanding the biology behind the observed racial disparities in survival among TNBC patients. There are relatively few studies that address the biological differences in TNBC among underserved populations, so this research addresses an important need.

The overall goal is noteworthy, and this research provides a good starting point. However, the interpretations and conclusions drawn from this research are overreaching for several reasons. First, the manuscript is overly dense with information, and as result, it lacks clarity on details. Second, there are inconsistencies throughout the manuscript with respect to what subjects and samples were utilized to draw these conclusions. Third, the methods utilized generate an abundance of data based on a relatively small number of patients. It was also unclear what subjects/samples were being utilized for the different experiments (discovery versus validation phase).

For example, IHC was done on tissue microarray samples from 31 WA and 26 BA according to Table 1 (although Figure 1 has 32 WA). Representative regions of interest (ROI) were selected from these TMAs, which appear to include a total of 47 BA and 51 WA samples. While the manuscript states that equal number and categories of ROIs were selected from tumors in BA and WA patients, it is unclear how this selection was done since the numbers don't reflect this. Furthermore, there were multiple ROIs selected from the same subject, but the manuscript doesn't state the number that came from each patient. Table 1 includes the number of patients in the different categories, while ROI samples are based on the number of TMA samples. Oddly, there is 1 BA subject who is positive for HER2, which doesn't make sense for TNBC patients. The text states that the WA and BA patients were not significantly different in mutational status for TP53 and PIK3CA (Table 1), yet these variables are not included in Table 1. Table 2 includes results from a Cox Proportional Hazards Analysis, but there is little explanation of these results. For example, the title of Table 2 could be more detailed on what is being modeled and some interpretation of the results (i.e. interpretation of the HRs) in the text would be beneficial. Also, matched studies typically use a matched analysis approach. Was this done, and if not, why? In Figure 2g and h, very small p-values were obtained, but no details are provided on how they were obtained. The observed differences do not appear to be that noticeable, and there is a large spread in the data. Figure 2f doesn't add much to this panel, as it is very difficult to see anything in these small panels. Figure 3 also lacks clarity. What does the Delta Prob represent? For example, in panel 3a) the value is 0.26, yet there is an indication (***) that there is a significant difference, so this must not be a P-value? This is not standard notation. The survival curves indicate increased risk for the BA-specific combined interaction scores for both races, not just BA patients. Is this race specific?

A spatial transcriptomic dataset was used to validate the proteins associated with the top 5 BA and WA cell-cell interactions identified. Protein names were converted to gene names, which were then queried in the spatial transcriptomic dataset. The transition from the original 20 IMC clusters to racially-specific 9 BA and 6 WA spatial cell-cell interactions, then to summed cell-cell interaction scores, then to proteins associated with the top 5 BA and WA cell-cell interactions, and finally to the 5 top gene queries, is difficult to follow. The manuscript would greatly benefit from a flowchart that summarizes these experiments and what samples/data are being used for each step. The manuscript then transitions to niche-specific gene signatures. Validation of the niches was done using both samples from an independent cohort, as well as from the original samples from the IMC analysis. It is unclear why the original samples are being used in the validation. Supplemental Tables 2 and 3 appear to be the same list of IMC antibodies, with the exception of 1-2 antibodies. Why are these both needed?

The manuscript would greatly benefit from improved organization and more clarity. In addition, the manuscript is overly dense with methods, data, results, figures, and tables. The progression from experiment to experiment is not completely clear, and often requires excessive searching between the methods, results and figures/tables to understand what was being done. Finally, given the many limitations, the conclusions drawn from this research could simply be an overinterpretation of the data. A more rigorous approach to validation of these findings is warranted.

(Remarks on code availability)

Version 1:

Reviewer comments:

Reviewer #1

(Remarks to the Author)

General Comments:

The revised manuscript by Zhu et al. demonstrates substantial improvement through the inclusion of new analytical approaches and additional data. The authors have thoroughly addressed prior reviewer feedback, resulting in a more rigorous and impactful study. Most notably, the integration of imaging mass cytometry (IMC)-based validation of 46 triple-negative breast cancer (TNBC) tumors from Roswell Park Comprehensive Cancer Center enhances the robustness and reproducibility of their findings.

A major strength of the revised manuscript is the shift from analyzing individual cell-cell interactions (CCIs) to identifying higher-order spatial “communities” composed of multiple, functionally related CCIs. These communities were validated across three independent clinical cohorts, demonstrating greater resilience to institutional and biological variability compared to the previously used BA Q1–Q5 and WA Q1–Q5 CCI categorizations.

The manuscript’s emphasis on racially distinct tumor microenvironments (TMEs) in TNBC is both timely and highly relevant. The updated analyses provide stronger support for the claim that spatial organization within the TME contributes to prognostic differences. Nevertheless, several suggestions remain for further refinement:

Remaining Concerns and Recommendations

1. Improve Accessibility of Complex Content

The manuscript contains dense and technically complex spatial biology concepts, which may overwhelm readers despite its scientific depth.

Suggestions:

- o Include a graphical abstract or schematic summarizing the analysis pipeline (e.g., IMC → Communities → ESGs → Prognosis → Validation).
- o Add a visual summary table outlining each cohort (e.g., BSW, Roswell, Georgia), the analysis performed (IMC, ST, GeoMX), key findings, and validation outcomes.
- o Reorganize the Discussion section to highlight biological insights—such as BA-niche and WA-niche characteristics and their therapeutic implications—rather than reiterating results. Additionally, consider briefly addressing the clinical utility of niche-specific signatures for patient stratification or trial design.
- o The discussion should also contextualize discrepancies with previous findings regarding the prevalence of exhausted T cells in Black patients (e.g., Yao et al., JNCI 2021, also from Roswell Park investigators). Clarify whether these differences are attributable to the more granular niche-based analysis used here.

2. Clarify the Added Prognostic Value of Spatial Context Over Cell Abundance

While the rebuttal and Supplementary Table 1 highlight that single-cell abundance alone lacks prognostic value, this critical point could be more explicitly presented.

Suggestions:

- o Introduce a dedicated subsection in the Results titled “Spatial Interactions Add Prognostic Value Beyond Cell Abundance.”
- o Provide a concise explanation of why spatial proximity—rather than mere immune cell presence—better predicts outcomes, particularly in BA niches.

3. Refine Interpretation of Immune Phenotypes (Suppl. Fig. 19)

The comparative analysis of BA/WA niche-enriched signature genes (ESGs) with inflamed/excluded/ignored immune phenotypes is insightful but remains correlative.

Suggestion:

- o In the Discussion, clarify that the observed alignment with known immune phenotypes is hypothesis-generating and warrants functional validation (e.g., assessment of immune signaling activity or cytokine profiling).

4. Address Potential Socioeconomic Confounders

The prognostic differences observed in WA-Comm1 between the Baylor and Roswell cohorts may be influenced by underlying socioeconomic factors, including food insecurity or access to care.

Suggestion:

- o If socioeconomic data are not available, include a statement acknowledging this limitation and discussing how such factors may contribute to cohort differences in immune function and outcomes.

Minor Comment

- The conclusion refers to unique immune “niches” in tumors from white patients. However, the data appear to reflect generalized immune infiltration rather than distinct spatial congregation into niches. Clarification of this distinction would improve the precision of the conclusions.

(Remarks on code availability)

Reviewer #2

(Remarks to the Author)

I co-reviewed this manuscript with one of the reviewers who provided the listed reports. This is part of the Nature Communications initiative to facilitate training in peer review and to provide appropriate recognition for Early Career

Researchers who co-review manuscripts.

(Remarks on code availability)

Reviewer #3

(Remarks to the Author)

I have reviewed the revision of the manuscript "Integrative spatial omics reveals distinct tumor-promoting multicellular niches and immunosuppressive mechanisms in Black American and White American patients with TNBC". In the first review round, I pointed out that the manuscript was lengthy and suggested to trim it down to the essentials. The manuscript is not shortened significantly, however, it is now written in a more precise manner. Especially the description of the cohorts is improved, and illustrations of the cohort and available data are included so it is easier to understand what has been done. I still think that the number of figures is excessive. Most of them are illustrative, however figure 3 and suppl. Figure 9 are very difficult to interpret and does not illustrate well the cell communities. I would consider revising these. It is not possible to appreciate the ROI segmentation into AOs for the geomx data in suppl. figure 2. My other concerns from the first review have been mostly addressed, either because changes have been made or that figures have been removed. I have no comments to the changes made to accommodate comments from the other reviewers.

(Remarks on code availability)

The code is available on GitHub. I lack competency in python, therefore, I can not review the quality of the code and the reproducibility.

Reviewer #4

(Remarks to the Author)

The authors have made extensive revisions to the manuscript, which have enhanced its clarity and quality. They have also satisfactorily addressed the concerns that I previously raised.

(Remarks on code availability)

Response to Review Comments

We thank the reviewers for an in depth and constructive review of our manuscript. Below we have addressed all the review concerns in a point-by-point manner. Importantly, in this revised manuscript we have included the following major changes/additions:

- 1) We have included imaging mass cytometry-based validation of 46 TNBC tumors obtained from Roswell Park Comprehensive Cancer Center. We have also used this dataset to generate additional Kaplan-Meier plots (see **Figures 3 c, d**) to verify the prognostic value of our findings.
- 2) We have used the cell-cell interactions that was reported originally using the imaging mass cytometry data to derive higher order communities. These higher order communities were universally verified across three clinical sample cohorts. Importantly, unlike cell-cell interactions (CCI) queries (BA Q1-Q5 and WA Q1-Q5) described in our original submission, communities composed of groups of CCIs were robust to intrinsic differences between institution-specific clinical cohorts. These validated communities were further used to generate the Extended Gene Signatures (ESGs) and prognostic analysis. The above additions have resulted in new Figure 3 and Tables 2 and 4

Point-by-point response to review comments

Reviewer #1 (Remarks to the Author):

1) The paper presents a study on triple-negative breast cancer (TNBC) that reveals distinct spatial cell-cell interactions between tumors from self-reported Black American and White American women. Despite using a racially balanced and clinically matched cohort, the study found no significant differences in overall survival rates or mutational statuses of key genes between the two groups. However, the spatial organization and interactions within the tumor microenvironment (TME) were racially distinct, with BA tumors showing more endothelial-macrophage-mesenchymal interactions and WA tumors displaying exhausted T-cell and neutrophil interactions. While the study is of importance and the findings are of interest there are significant concerns with the interpretation of the data by the authors in addition to the clinical translation and broader impact being limited.

Response: We are pleased to note that the reviewer finds our work to be important and findings of interest. In the revised manuscript, we have modified the interpretations of the data for better clarity. Importantly, our study for the first time describes unique tumor microenvironment (TME) niche associated with TNBC in Black American (BA) patients. Our work lays the foundation for future clinical translation aimed at developing therapeutic strategies that can effectively target the tumor-endothelial-macrophage niche in TNBC. In addition, the BA-niche signature could serve as a tissue-based prognostic signature. We expect these to benefit the BA TNBC patients and thus reduce the disparities in TNBC outcomes. We have included a paragraph describing the above in the Discussion section that reads “These observations underscore the potential of race-associated cell-cell interactions as strong prognostic markers of survival in TNBC. The ability to identify specific cellular communities and their interactions within the TME offers new avenues for understanding how these interactions influence disease progression and therapy responses in a race-

dependent manner. As illustrated by the BA-Community-1 example, race-specific interactions in the TME may influence the long-term clinical outcomes of patients, indicating the need for tailored approaches to treatment that account for these differences. Our results are in line with recent research that demonstrates the role of race and ethnicity in shaping immunotherapy responses in breast cancer patients¹³, further validating the significance of these findings for the development of personalized therapies”.

2) This manuscript used regions of interest evaluated from the tumor samples using imaging mass cytometry and selected by a breast pathologist. This appears to be done based on the location of the ROI and the terms “immune rich” vs. “immune poor.” Were those terms evaluated using antibody-based staining of immune cell markers in the tumor microenvironment (TME) or just by an H&E evaluation of cells that appear to be immune cells? Did the authors have a second pathologist evaluate the sections for standardization? This is critical as these findings were not completely recapitulated using 10X analysis of whole tumors.

Response: We have included new **Supplementary Figure 3** where we have stained the ROIs using CD45 antibody to quantify the total immune cells. Our analysis shows that immune rich ROIs defined by the pathologist using H&E images (TCIR and TPIR) had significantly higher number of immune cells compared to immune poor (TCIP and TPIP) regions. Further, we demonstrate high degree of concordance in the data obtained from the multiple ROIs within each tissue. We also combined the two ROIs in our analysis. We were also able to replicate the IMC-based findings using 10X ST analysis of the whole tumors. In addition, we have also evaluated the H&E based ROI selection with the help of two breast pathologists and obtained 100 % agreement in the stratification results. To address the reviewers concern on the protocol that was followed for ROI selection, we have included the following paragraph in the “Methods Section” in the revised manuscript. “TMA cores were at least 3mm in diameter. Tumor periphery was defined as within 1mm of tumor perimeter when identifiable. Tumor center was defined as greater than 1mm from the tumor perimeter. Stromal tumor infiltrating lymphocytes were determined using the International TILs Working Group 2014 guidelines (PMC6267863) on H&E slides, following completion of available tutorials at tilsinbreastcancer.org. Less than 10% was considered immune poor and greater than 50% was considered immune rich. A single pathologist (JRA) selected all ROIs at a single sitting. These ROI were verified by a second reviewer (PBD) with 100% agreement”.

3) The initial evaluation of cell clusters in figure 1B is somewhat difficult to assess. For instance, cluster 12 is composed of cells expressing PD-L1, CD16 and CD163, which are found on a number of different cell types including macrophages, monocytes, NK cells and some dendritic cells. Thus, what is this cluster and is it composed of all of these cells, or a subset of them. How do the authors assess cellular interactions for these clusters when the cells that are potentially present are this broad? Similarly, although B cells have been clearly found in TNBC samples (Conte et al. EBioMedicine 2024), B cells are not clustered from the TMAs. Why is that? It would be helpful for the authors to take one of the TMAs and stain this with standard antibodies (CD4, CD8, CD19, CD14, CD11c) and compare the cells identified with the cell clusters identified by mass cytometry.

Response: The reviewer’s observation – that there exists in some cases combination of markers in a cluster – could be attributed to the limitations in the current cell segmentation methods, which

could find two adjacent cell-types' markers in one cluster. But we note that this happened in 4/20 clusters and these 4 clusters contain only 2-3 interacting cell types (thus not "broad" as reviewer claims). In the revised manuscript, we greatly improved the interpretability by assigning intuitive names to all 20 clusters, such as Exhausted Cytotoxic T cells, T-helper cells, etc, as illustrated in **Figure 1d**.

To address this issue more robustly, in the revised manuscript, we have shifted our focus from analyzing individual cellular interactions (CCIs) to a community-based approach. In this approach, we group related CCIs into communities of interacting cell types, which simplifies the analysis and provides a clearer biological context. These communities were validated across three independent IMC datasets (**Figure 3**) and have been shown to represent consistent patterns of cell interactions. This community-based analysis is conforming to the standard analysis procedure in other IMC papers (refer PMIDs 31959985, 36725934). In addition, the BA and WA community verified using multiple IMC data was further examined for its robustness using three independent 10X Visium ST datasets. BA community 1 and WA community 1 that passed this rigorous multi-step validation procedure were used to define the Extended Gene Signature (ESGs) and downstream ligand receptor interactions.

In response to this reviewer's concern about B cells in TNBC samples, we have measured CD20 as a B cell marker using IMC in our new validation set from Roswell Park cohort. Our data shows that CD20 sits in a separate WA-community-3 with M2 macrophages (**Supplementary Figure 9**).

4. This reviewer is quite confused on how the Giotto spatial network function and Delaunay network can be used for evaluating differences between patients based on race given the differences in cellularity across the samples. Giotto is typically used to infer cellular relationships by using a triangulated grid to infer cellular interactions, so it is unclear how the authors are comparing specimens with significant immune infiltration versus those without this. Presumably the latter will have far fewer cellular networks but could still have productive interactions between cells. Is this being evaluated? How do the authors differentiate racial differences in CCI between racial differences in cellular heterogeneity or abundance? This reviewer strongly believes that the best way to do this analysis would be to evaluate amongst the FoxP3+ population how many of these interact with DCs (for example) independent of the total number of FoxP3+ cell interactions. If that were done, are there still different interactions between the patient groups? It's not clear to this reviewer that the Giotto spatial evaluation can be used to compare interactions across tissues but should be used to infer potential interactions within a tissue. This was done in Figure 2H but not done for any of the other purported cellular interactions. Finally, it's not clear to this reviewer why the authors believe that the CD163+ cells are macrophages and not blood monocytes interacting with endothelial cells as part of their migratory activity.

Response: Giotto is a previously validated method for cell-cell interactions (CCIs) (refer to Genome Biology PMID: 33685491). In the original paper, we have demonstrated that Giotto can recapitulate CCIs described in from MIBI platform which is similar to IMC (PMID: 30193111). We have now included **Supplementary Figure 5** where we describe how Giotto calculates the cell-cell interaction score for 9 different possible spatial arrangements of interacting cell types. These examples illustrate Giotto's ability to detect patterns across a broad spectrum of scenarios, including cases with unbalanced cell type distributions, variations in abundance, and heterogeneity, as requested by the reviewer.

Regarding comparisons between samples, our prior publication successfully demonstrated Giotto's ability to analyze interactions across tumor samples, identifying two broad classes of spatial arrangements: compartmentalized versus intermixed (PMID: 30193111). Briefly, Giotto enables meaningful comparisons between samples despite differences in immune cell content by generating an empirically derived background. This is achieved by randomly shuffling the spatial positions of cell types within each sample and computing the expected number of interactions under randomized conditions. By comparing observed interactions against this sample-specific background, Giotto calculates an enrichment score that accounts for inherent cellular heterogeneity and varying cell abundances across samples. Indeed, Giotto can accurately detect interactions even with an eightfold difference in cell abundance between interacting cell types (see Case 2 in **Supplementary Figure 5a**). Additionally, Giotto reliably identifies scenarios with no interaction (**Supplementary Figure 5c**) and cases demonstrating interaction depletion (**Supplementary Figure 5b**).

We consider CD163+ cells are macrophages based on prior publications (PMCID 9931585). In addition, a recent study demonstrates that nanobody-based immunotracer targeting CD163 is more specific for macrophages compared to clinically used anti-CD206 macrophage tracers (DOI: 10.1073/pnas.240966812).

Regarding the reviewer's suggested measure—the proportion of a specific cell type (e.g., FoxP3 cells) interacting with another—we note that directly comparing these proportion values across different samples is not meaningful unless each proportion is first tested for significance within its own sample. Instead, it is the resulting p-values from these tests that can be meaningfully

Revision Figure 1: Comparison of CCI p-values between those derived by Giotto and those derived by computing proportions of cell types that are interacting.

compared across samples. To assess this, we conducted a small-scale analysis using 1 ROI comparing p-values derived from proportions of a cell type interacting with another (Reviewer's method) to p-values obtained using total number of interactions between 2 cell types (Giotto's method). Our results (see **Revision Figure 1**) demonstrated a high correlation between these two approaches (Pearson correlation = 0.95), supporting the validity of Giotto's method. Across all of the ROIs, the Pearson correlation remains at 0.955 between the two methods, indicating consistent identification of CCIs regardless of the chosen method. Again, this is because of the use of sample-specific randomized background in Giotto, a critical feature of its analytical framework.

5. The robustness of the spatial interactions in the current data set is unclear to this reviewer. Thus, is there significantly more prognostic information obtained from the regional assessment of cell expression compared to overall single cell expression from the sample. The authors insinuate that significant cell-cell interactions are driving the poor outcomes in different racial groups. However, it is not clear to this reviewer whether this is true or that the presence of immunosuppressive macrophages or regulatory T cells are driving the outcome independent of

the cells they interact with. Can the authors confirm that there is significantly more prognostic information obtained from performing spatial assessment compared to only single cell assessment? If this were true, one would infer that the authors could demonstrate downstream changes in cells that are interacting with other cells from the spatial transcriptomic data. Was this true?

Response: We have demonstrated the robustness of the CCIs by carrying out leave-two patient-out cross validations. In addition, we have validated our findings in independent tissue cohorts using both IMC and 10X Visium ST. The objective of the manuscript is to demonstrate the unique CCI-derived communities in the tumor microenvironment of TNBC tissue derived from BA vs WA patients. BA patients have aggressive tumors with poor clinical outcome and more often than not they are resistant to standard of care treatments that are designed using data obtained from WA patients. Of late, there have been studies published in high profile journals showing that the TNBC disparities are associated with biological differences.

In response to the reviewer's question on whether there is more prognostic information obtained from performing spatial assessment compared to only single cell assessment, we have included results of new analysis in **Supplementary Table 1** showing lack of prognostic information using only single cell assessments. Specifically, our results demonstrate that only 3 of 20 single-cell clusters are significantly different in abundance between BA and WA, but none of these are associated with overall survival (i.e., non-significant HR values). There are only 2 very weakly significant clusters associated with overall survival (clusters 16 and 17), but these are not race-specific. Thus, we conclude that single-cell abundances do not contribute to race-specific overall survival. In contrast, **Figure 3** and **Supplementary Figure 10** show the prognostic value of community-associated gene signatures in a race dependent and independent manner.

Regarding downstream changes in interacting cells, our gene-set enrichments in **Figure 5 e-f**, and ligand-receptors in **Figure 6a**, made from race-specific ESGs are showing these downstream altered pathways.

6. For the confirmatory data shown in figure 7, the authors need to show all of the extended signature gene groups. Did the additional platforms verify the BA-Q1 or Q2 ESG's? Similarly, did they verify the WA-Q1, Q4 ESG's? If they did not, what is the author's interpretation of those findings? Does that question the validity of these ESG's as being identified differently in the different racial groups with triple negative breast cancer? And if the authors argue that the differences are due to ROI vs. whole tumor assessment, isn't the whole tumor evaluation more robust? This suggests to this reviewer that there is significant bias in the assessment of the ROIs using imaging mass cytometry

Response: In the revised manuscript, instead of focusing on CCIs, we have used the IMC data to define **BA and WA cell communities** which are high order spatial cell-cell interactions. We have verified these communities across three IMC datasets namely BSW-Discovery, Roswell Park-Validation and BSW2-Validation. We have then used the community signature that was validated across the three data sets (i.e., BA community 1 and WA community 1) to further verify using three independent 10X Visium ST data sets namely Bassouni *et al*, BSW2-Validation and Georgia-Validation cohorts. The community signature that was verified using the above-mentioned steps were then used to create the ESGs. Interestingly, these ESGs are highly similar to the ones we had earlier reported using cell-cell interactions, verifying their robustness.

With the introduction of cell communities, we no longer have BA-Q1 through Q5 or WA-Q1 through Q5, resulting in a simpler analysis. We note that these individual queries are validated within their respective communities (BA community 1 and WA community 1) using the datasets mentioned above.

Regarding the concern about the ROI versus whole tumor assessment, we acknowledge that whole tumor assessments provide a broader, more comprehensive view of tumor heterogeneity. However, ROI-based analysis allows for a more focused exploration of specific regions of interest within the tumor microenvironment where critical cell-cell interactions are occurring. While whole tumor evaluation is more robust in some aspects, we argue that ROI-based analysis offers complementary insights that can refine our understanding of the spatial organization of immune cells in the TME. Additionally, ROI selection was done with rigorous standardization and validation across datasets, ensuring that bias was minimized.

7. There was almost a significant difference in the number of low-grade tumors in the two racial groups. The authors need to perform their analysis removing the low-grade tumors and only analyzing the grade three tumors. Were the findings still significant once these lower grade tumors were removed? Similarly the metastatic lesions should be removed from the analysis given differences in the immune response in metastatic lesions. Why would the authors choose not to exclude the case with HER2 positive disease amongst Black patients, since this study focuses on TNBC and given significant differences in clinical, immune and molecular characteristics between TNBC and HER2 positive, the exclusion of this case may be warranted? Suggest grouping stage into 1,2,3,4 rather than dividing into subgroups within each stage. This is for simplicity, and also, clinically the subgroups do not add to the significance of the analysis

Response: To address the concern of the reviewer, in the revised manuscript, we have included **Supplementary Figure 6** showing that the results remain unchanged even after removing the low-grade samples. We did not use any metastatic lesions for the analysis. The Her2 sample has been re-examined using FISH and confirmed to be TNBC. We have also grouped the samples based on this reviewer's suggestion in **Tables 1 and 2** in the revised manuscript.

8. The author should analyze the triple negative breast cancer groupings with the more granular assessment of intrinsic genes that separate out specific tumors as described in the discussion and published in the literature. It would then be helpful to try to ascertain whether there were specific features of the racially profiled tumors that were associated with basal-like immunomodulatory or the mesenchymal-like subtype. Finally, it would be quite helpful for the robustness of this manuscript if the authors were able to confirm cellular interactions leading to changes in gene transcription. Can they for instance demonstrate differences in macrophage activation based on their interaction with CD 31 expressing endothelial cells?

Response: In response to the reviewer's concern, we performed subtyping of triple-negative breast cancer (TNBC) using Nanostring GeoMx gene expression profiles from 97 regions of interest (ROIs) in the BSW-Discovery cohort of patients. The subtypes identified were BL1/2 (Basal-like), IM (Immunomodulatory), LAR (Luminal AR), M (Mesenchymal), and MSL (Mesenchymal Stem-like). The proportions of patients in each subtype for both BA and WA groups are de-

picted in the accompanying pie charts (**Revision Figure 2**). Notably, BA exhibited a higher proportion of BL1/2 compared to WA, while WA demonstrated a higher proportion of IM/M and IM/MSL subtypes than BA. These findings suggest a more immunogenic tumor microenvironment in WA patients.

The hybrid classification (IM/M or IM/MSL) highlights a limitation in existing TNBC classifications, as many IM niches exhibit a mixture of gene signatures from other subtypes. We believe this deficiency in existing TNBC classification resulted in the erroneous observation that WA patients display a higher mesenchymal content (M or MSL) than BA. To remedy this situation, we have adopted a specific immunosubtyping method developed in PMID 31451770, which classified TNBC tumors as macrophage-enriched subtype (MES) or neutrophil-enriched subtype (NES). Using this classification we saw that BA patients are predominantly MES while WA patients are predominantly NES subtype (**Revision Figure 3**). Because MES indicates higher activity of EMT, and adopts a spindle-like and mesenchymal-like morphology, the finding translates to BA tumors being more mesenchymal like than WA tumors, correctly validating our BA and WA characterizations.

Revision Figure 2: TNBC Subtyping performed on BSW-Discovery cohort, based on Nanostring data. For BL1, BL2, LAR, M, MSL subtypes, we used PanCK+ compartment for subtyping. For IM, we used CD45+ compartment. Each patient is assigned to the epithelial subtype that has the highest positive subtype signature score. Separately, a patient is assigned to IM subtype if its IM signature score (average z-score) exceeds 0.25.

Revision Figure 3: TNBC MES and NES subtyping performed on BSW-Discovery cohort, based on Nanostring data. PanCK compartment was used.

Regarding cellular interactions leading to changes in gene transcription, we isolated ST spots with and without macrophage-endothelial interactions and compared their differentially expressed genes to study how interactions affect transcription. Interaction induced processes such as cell-cell communication and receptor signaling (**Revision Table 1** below). Spots with interactions also had more CD163+ macrophages—an M2 activation marker—which uniquely engaged in myeloid leukocyte activation ($P=3.1e-5$) and antigen processing and presentation ($P=1e-6$), compared to CD163- CD68+ macrophages.

Comparing CD163+, CD68+, CD31+ subset with CD163+, CD68+, CD31- subset. GO-enrichment	-log₁₀(Pvalue) (Average of Bassiouni et al, BSW, Georgia ST datasets)
GO:0001568-blood vessel development	32.86
GO:0007154-cell communication	11.09
GO:0050896-response to stimulus	13.01
GO:0007166-cell surface receptor signaling pathway	12.91
GO:0071363-cellular response to growth factor stimulus	6.15

Revision Table 1: Comparing CD163+ CD68+ CD31+ ST spots with CD163+ CD68+ CD31- ST spots

9. Table 1 describes that majority of cases (22 vs 29) did not receive chemotherapy. Given majority of cases are TNBC, this reviewer finds it odd that patients did not receive chemotherapy as part of their treatment paradigm. Did the authors mean to say that the tissue was analyzed prior to receipt of chemotherapy? If so, suggest clarification as chemotherapy would drastically affect outcomes in this setting. Finally, for the survival analysis, did any of the patients receive immune checkpoint inhibitor therapy, which is standard for neoadjuvant therapy for patients with TNBC and would confound the survival interpretation.

Response: In our BSW-Discovery data set, four of the patients (2 each of BA and WA) received neoadjuvant chemotherapy. All the remaining patients were treatment naive when the tumors were resected at surgery. These surgically resected samples were used for the discovery analysis. Later, in the adjuvant setting, all the patients were treated at the same hospital using the standard of care regimens. None of them received immune checkpoint therapy. Two patients were enrolled in a SWOG clinical trial. This is now described in Methods section under the subheading “*TNBC patient tissue microarrays*”

10. The authors highlight BMI differences between Blacks and Whites; although this did not translate into survival differences in Table 2, it would still be interesting to look at the interaction between BMI and the genomic findings highlighted. Also, this should be adjusted for in survival analysis, given BMI is a known contributing factor to disparities in outcomes as previously published

Response: We have included BMI in the multivariable analysis in **Table 3**. Our results show that BA-communities but not WA communities are significantly associated with poorer outcome even after adjusting for age, race, BMI and stage in the multivariable analysis (BA: HR 7.69, $P = 0.021$, WA: 1.27, $P = 0.368$).

Minor:

1. The authors indicate that p53 and PIK3CA status did not differ between the cohorts and indicate that this is in Table 1 but this information is found in Table 2. It would also be important to show these data as VAF fraction and mutation incidence rather than HR as indicated in the table.

Response: We apologize for the typographic error. This information is now in Table 3 in the revised manuscript. We don't have information on the VAF fraction and mutation incidence for these genes in the data set.

2. This reviewer was not able to identify the statistical tests used for comparisons in figures 2F, G and H. This information should be provided in the legend to the figure.

Response: We used Mann-Whitney test which is now indicated in the legends in the revised manuscript.

3. It is assumed by this reviewer that the columns in figures 4C and E represent the different subgrouping of interactions shown in figure 4B. Is this correct?

Response: In the revised manuscript, we no longer having BA-Q1-Q5 or WA-Q1-Q5. These are now replaced by BA-community-1 and WA-community-1 that describe the overall cell-cell interaction groups observed in BA and WA TNBC tumors, respectively. Given this, Figure 4C and Supplementary Figure 12 now describe the overlay of genes associated with BA and WA community-1 respectively in 10 BA and 10 WA TNBC tumors reported in the racially inclusive publicly available Bassiouni et al ST dataset.

4. It was somewhat surprising that the increased cell to cell interactions in the cohort of white patients was associated with an inferior prognosis given the presence of adaptive immune cells in those samples. This has also been more associated with black patients with TNBC (Yao S et al. J Natl Cancer Inst. 2021 Aug 2;113(8):1036-1043. doi: 10.1093/jnci/djaa215. PMID: 33395700; PMCID: PMC8328978.). A more elaborate discussion regarding this would be helpful Did any of these patients receive immune checkpoint inhibitor therapy? Was this treatment associated with a different outcome?

Response: In the revised manuscript, we define high order communities using the cell-cell interactions that were reported earlier. The BA community signature but not the WA community signature was associated with inferior prognosis across both BA and WA TNBC patients. As described in the manuscript, WA TNBC contain exhausted T cells in their WA-community signature, which explain the inferior prognosis described above. None of the patients in our study received immune checkpoint therapy. However, as alluded to by this reviewer, based on the findings of exhausted T cells in the WA patients, we believe they may be more responsive to immunotherapy, a premise that needs to be further examined in an independent study.

We also expanded the Discussion: "The significance of spatial datasets in comprehending the roles of immune cell types is also underscored by our findings, as the abundances of cell types or the deconvoluted cell proportions from bulk RNAseq analyses might not accurately reflect the intricate nature of the tumor microenvironment in patients with BA and WA. Earlier research on

the prevalence of T cells and Treg cells among different races, as well as their significance, would probably require a reevaluation using spatial technologies. The new spatial immunophenotypes described by Gruosso et al and Hammerl et al are a step towards this direction. Our work further expanded this knowledge by including additional cell-cell interactions and community network as features of race-specific TNBC tumors.”

5. Typo noted in results section “However, WA patients tend to have a higher risk for poor OS when stratified by WA-associated interactions (see WA-CCI * WA race in Table 2, HR 1.46)”. HR in table is 1.56. please specify which is correct HR.

Response: We no longer describe association between overall survival and CCI in the revised manuscript.

6. Without additional information, it is highly speculative and discussion section for the authors to argue that racial differences in the immune landscape of triple negative tumors should be investigated using genetically engineered mice. It is entirely unclear to this reviewer how these mice can inform on “racial differences” in the immune landscape without additional information regarding what processes are driving this finding.

Response: As requested by this reviewer, we have removed the section in Discussion that argues that racial differences in the immune landscape of triple negative tumors should be investigated using genetically engineered mice.

Reviewer #2 (Remarks to the Author):

Reviewer #3 (Remarks to the Author):

1. The manuscript titled “Integrative spatial omics reveals distinct tumor-promoting multicellular niches and immunosuppressive mechanisms in Black American and White American patients with TNBC” is a mainly well written and thorough paper on a highly relevant topic that has received too little attention previously. Racial disparities in the immune microenvironment of triple negative breast cancer can explain some of the differences in outcome between black and white women. In this manuscript the authors address this topic utilizing several methods exploring the immune tumor microenvironment, including imaging mass cytometry, immunofluorescence and different methods of spatial transcriptomics.

This manuscript is comprehensive and includes a lot of analyses and results, to a point where it is overwhelming to read, and it is difficult for a reader to keep track of the main message. I

would highly recommend thinking about what MUST be included to keep the most important messages from drowning and try to strip the manuscript of unnecessary analyses. Figures 1-4 should be included, however, from there, I would recommend making only 1-2 figures with the most important results and also try to limit the number of supplementary figures. This said, the manuscript can absolutely become publishable if rewritten and shortened.

Response: The reviewer praised the manuscript as well written and thorough, highlighting its relevance that has been underappreciated. They suggested that with rewriting and shortening, the paper could be published. We have revised the manuscript following the reviewer's guidance to improve clarity. Despite our intention to shorten, we were unable to do so at this juncture to address the concerns of the other reviewers of the manuscript. We have however moved one of the main figures (**Figure 5**) to the Supplementary (**Supplementary Figure 5**). We have also simplified the interpretation of the results by combining the different cell-cell interactions in BA and WA into a single higher order community. This has also resulted in reduction of individual figure panels (see **Figures 6 and 7** in the revised manuscript). Once all the reviewers accept the manuscript, we will be happy to trim the text and figures during the final editing phase.

2. The introduction is well written and the material and methods sections include sufficient detail. The figures are not always informative, especially figure 6 and 7.

Response: In the revised manuscript, these now refer to **Figures 5 and 6**. **Figure 5** describes extended gene signatures associated with BA and WA community highlighting the involvement of additional cell types. **Figure 6** examines the BA and WA ESGs to determine the downstream signaling denoted by ligand-receptor pairs.

In addition I have some minor comments:

- It needs to be explained whether IMC and GeoMx is run on different slides. If they are run on different slides, are the ROIs for IMC and GeoMx the same/in the same area or different areas?

Response: This has been clarified in the "Methods Section" in the revised manuscript under the subsection entitled "**Region of Interest (ROI) selection and segmentation**"

- I would recommend including an image of an GeoMX ROI showing segmentation into the AOs.

Response: Supplementary Figure 2 shows all the ROIs across all the tumor samples analyzed. Within each ROI, the AOs are marked by PanCK (green color, epithelial cells) and CD45 (red color, immune cells).

- Suppl table 1: MSC (abbreviation should be explained). Mesenchymal – one "s" to many

Response: The above mentioned table has been removed from the revised manuscript.

- Figure 1D – color legend needs to be added

Response: This is now Figure 1f in the revised manuscript. The Z score scale bar is now included.

- Supplementary figure 3A: the colors used in the dendrogram needs to be explained

Response: This is now Supplementary Figure 4. Since the colors in the original dendrogram have no meaning, we have made them monochromatic in the revised manuscript.

o It could be an idea to cluster the data using the relative abundance of immune cells only. In that way, the different abundance of tumor cells would not influence the clustering and it might uncover differences between BA and WA? (remove KIFC1, Ki67, PLK1, pHH3, PanCK, VEGF, ECad, HIF1 a, AR, PLK1)

Response: In response to the reviewer’s suggestion, we recalculated the proportion of cells within each immune compartment, rather than considering the total cell count. Subsequently, we compared the proportions between the BA and WA patient groups. This analysis revealed three additional clusters whose abundance showed a significant association with race (See Table below). Conversely, two clusters that previously exhibited a race-related association, based on total cell proportions, no longer demonstrated such a relationship (See **Revision Table 2** below). Collectively, these findings reinforce our initial observation that cell-cell interactions (CCIs) exhibit stronger race-specific associations than the overall frequencies of clusters.

Cluster	Total cells per ROI, BA vs. WA (P-value)	Compartment-specific cells, BA vs. WA (P-value)
1a M2 Macrophages		
2a Tumor cells	0.004	0.0074
3a Dendritic cells		
4a Epithelial-Mesenchymal cells		
5a Tolerogenic Dendritic cells		0.04996
6a Hypoxia		
7a Exhausted Cytotoxic T cells		
8a Proliferating Tumor cells		
9a Cytotoxic cells in Hypoxia		0.00532
10a Myeloid cells		
11a Helper & Cytotoxic T-cells		
12a M2 Macrophages		
13a Endothelial-Mesenchymal ce	0.008	
14a Naïve and Memory T-cells	0.058	
15a T-Helper cells		
16a Angiogenic Tumor cells		
17a Proliferating Cytotoxic cells		
18a Proliferating Cells		
19a Immunosuppressive Tumor c	0.009	0.052
20a Naïve T cells		0.0456
T-test (two-tailed)		

Revision Table 2: Cluster abundance comparisons between BA and WA populations. Column 2: cluster abundance as proportions of total cells per ROI. Column 3: cluster abundance as proportions of compartment-specific cells per ROI. P-values indicate t-test statistical significance. Only clusters exhibiting significant difference between BA and WA are shown.

- Figure 2F – the Zoom for the selected area should preferably be larger

Response: This has been included in the revised manuscript.

- Figure 3E: Need to annotate that the 5 columns represent the 5 different queries

Response: The revised manuscript no longer focuses on the individual cell-cell interaction queries. Figure 3e has been deleted from the revised manuscript.

- Figure 5 could be a supplementary figure

Response: Yes we agree. Spatial overlay of the WA community-1 is now in Supplementary Figure 12.

- Figure 6 and figure 7 need to be better explained and justified.

Response: In the revised manuscript, these now refer to **Figures 5 and 6**. **Figure 5** describes extended gene signatures associated with BA and WA community highlighting the involvement of additional cell types. Figure 6 examines the BA and WA ESGs to determine the downstream signaling denoted by ligand-receptor pairs. In the revised text, **Figure 5** is now described under the subsection named “Niche-specific differential expression analysis reveals additional players associated with BA and WA-tumor associated multicellular niches”. Similarly, **Figure 6** is prefaced by the sentence “To investigate the drivers within the racially distinct spatial environments, we used ESGs, as described earlier, to extract ligand-receptor interactions within the BA and WA niches, revealing a unique set of ligand-receptor pairs”

3. The code is accessible, however I'm not sure that it will be enough to reproduce by others.

Response: We have made sure that the codes are accessible in GitHub and enough for others to replicate our analysis.

Reviewer #4 (Remarks to the Author):

1. In the manuscript entitled “Integrative spatial omics reveals distinct tumor-promoting multicellular niches and immunosuppressive mechanisms in Black American and White American patients with TNBC”, the authors seek to identify the biological mechanisms that potentially explain why Black American women with triple negative breast cancer (TNBC) have worse prognosis compared to White American women. They utilize a spatial omics approach

that integrates spatial single-cell level immunohistochemistry and spatial transcriptomics data to identify race-specific niches that potentially correlate with overall patient survival. By integrating these different technologies, this research provides a more comprehensive approach that could lead to new insight in understanding the biology behind the observed racial disparities in survival among TNBC patients. There are relatively few studies that address the biological differences in TNBC among underserved populations, so this research addresses an important need.

Response: We are happy to note that this reviewer highlights the fact that there are relatively few studies that address the biological differences in TNBC among underserved populations, so this research addresses an important need.

2. The overall goal is noteworthy, and this research provides a good starting point. However, the interpretations and conclusions drawn from this research are overreaching for several reasons. First, the manuscript is overly dense with information, and as result, it lacks clarity on details. Second, there are inconsistencies throughout the manuscript with respect to what subjects and samples were utilized to draw these conclusions. Third, the methods utilized generate an abundance of data based on a relatively small number of patients. It was also unclear what subjects/samples were being utilized for the different experiments (discovery versus validation phase).

Response: We thank the reviewer for this pertinent question. In response to the review concern we have included the following in the revised manuscript

1) Figure 1a describes the different sample cohorts and the technologies used for collecting the data. 2) Figure 1b includes a flow chart showing the different analysis that were performed. c) We have included a new cohort of 46 TNBC patients containing 15 BA and 31 WA samples from Roswell Park Comprehensive Cancer Center for validation of IMC results. We do not believe that we are over-interpreting results based on a limited sample size. Our IMC findings use 113 TNBC tumors and 10X ST Visium findings uses data from 39 TNBC patients.

3. For example, IHC was done on tissue microarray samples from 31 WA and 26 BA according to Table 1 (although Figure 1 has 32 WA). Representative regions of interest (ROI) were selected from these TMAs, which appear to include a total of 47 BA and 51 WA samples. While the manuscript states that equal number and categories of ROIs were selected from tumors in BA and WA patients, it is unclear how this selection was done since the numbers don't reflect this. Furthermore, there were multiple ROIs selected from the same subject, but the manuscript doesn't state the number that came from each patient. Table 1 includes the number of patients in the different categories, while ROI samples are based on the number of TMA samples.

Response: We have included the following two statements in the Methods section to address the reviewer's concerns. "For IMC analysis, tissues from 31 WA and 26 BA patients were used. A total of 98 ROIs were examined that included 47 ROIs from BA patients, and 51 ROIs from WA patients. This included one BA patient with 3 ROIs, 19 BA patients with 2 ROIs and 6 BA patients with 1 ROI tissues. For WA, we had 20 patients with 2 ROIs and 11 patients with 1 ROI". "The following guidelines were used to select the ROIs.

TMA cores were at least 3mm in diameter. Tumor periphery was defined as within 1mm of tumor perimeter when identifiable. Tumor center was defined as greater than 1mm from the

tumor perimeter. Stromal tumor infiltrating lymphocytes were determined using the International TILs Working Group 2014 guidelines (PMC6267863) on H&E slides, following completion of available tutorials at tilsinbreastcancer.org. Less than 10% was considered immune poor and greater than 50% was considered immune rich. A single pathologist (JRA) selected all ROIs at a single sitting. These ROI were verified by a second reviewer (PBD) with 100% agreement”.

4. Oddly, there is 1 BA subject who is positive for HER2, which doesn't make sense for TNBC patients. The text states that the WA and BA patients were not significantly different in mutational status for TP53 and PIK3CA (Table 1), yet these variables are not included in Table 1.

Response: Re-analyses of the HER2 tissue specimen using FISH confirmed it to be TNBC. We have made this change and included information on TP53 and PIK3CA status in revised Table 1.

5. Table 2 includes results from a Cox Proportional Hazards Analysis, but there is little explanation of these results. For example, the title of Table 2 could be more detailed on what is being modeled and some interpretation of the results (i.e. interpretation of the HRs) in the text would be beneficial. Also, matched studies typically use a matched analysis approach. Was this done, and if not, why? In Figure 2g and h, very small p-values were obtained, but no details are provided on how they were obtained. The observed differences do not appear to be that noticeable, and there is a large spread in the data. Figure 2f doesn't add much to this panel, as it is very difficult to see anything in these small panels.

Response: Thanks for this question. We have included additional details to describe the Cox Proportional Hazards Analysis and moved Figure 2f to the supplementary. We have also described details of the Cox Proportional model in greater detail in the Methods in the main text.

In response to the reviewer's comment regarding “matched studies using a matched analysis approach,” we would like to clarify that we made every effort to match the subjects in this study for age, stage, and race. To ensure the appropriate statistical model was used, we considered applying a stratified Cox model, incorporating the main effect (CCI) and stratifying by patient race, as recommended in the literature (PMID: 26712591).

In addition to the "stratified model," we also explored "separate models" and "interaction models," following the approaches outlined in the presentations by David Rocke (<https://dmrocke.ucdavis.edu/Class/EPI204-Spring-2021/Lecture13StratifiedCoxModels.pdf>). After comparing the results from all models, we did not observe significant differences in the derived P-values across the models in our dataset (**Revision Table 3**). Therefore, we chose not to pursue the stratified Cox model further. However, we fully acknowledge the reviewer's point that a stratified Cox model (i.e., for a matched study) is also a valid approach. We have included the AAscore results derived from the various models mentioned above in the following section for reference.

simple model						
cox<-coxph(Surv(time, status)~AAscore+race+stage+bmi+grade, data=x)						
	coef	exp(coef)	se(coef)	z	Pr (> z)	Pr (>z)
AAscore	3.32928	27.91811	1.47118	2.263	0.0236	0.0118
raceEA	0.39663	1.48681	0.88577	0.448	0.6543	0.3271
stage	0.89974	2.45896	0.53688	1.676	0.0938	0.0469
bmi	0.04557	1.04663	0.05216	0.874	0.3822	0.1911
grade	-0.50109	0.60587	0.25206	-1.988	0.0468	0.4766
subset model						
cox<-coxph(Surv(time, status)~AAscore+bmi+grade+stage, subset=(race == "AA"), data=x)						
	coef	exp(coef)	se(coef)	z	Pr (> z)	Pr (>z)
AAscore	4.456556	86.190125	2.392309	1.863	0.0625	0.0312
bmi	-0.002326	0.997676	0.078181	-0.03	0.9763	0.0118
grade	-0.347254	0.706626	0.421039	-0.825	0.4095	0.2952
stage	2.67435	14.502922	1.381602	1.936	0.0529	0.0264
stratified model						
cox<-coxph(Surv(time, status)~AAscore+stage+bmi+grade+strata(race), data = x)						
	coef	exp(coef)	se(coef)	z	Pr (> z)	Pr (>z)
AAscore	3.31947	27.64561	1.46303	2.269	0.0233	0.0116
stage	0.95427	2.59678	0.57038	1.673	0.0943	0.0471
bmi	0.04039	1.04122	0.0512	0.789	0.4302	0.2151
grade	-0.46135	0.63043	0.24705	-1.867	0.0618	0.4691
interaction model						
cox<-coxph(Surv(time, status)~stage+bmi+grade+AAscore*race, data = x)						
	coef	exp(coef)	se(coef)	z	Pr (> z)	Pr (>z)
stage	0.89157	2.43896	0.53988	1.651	0.0987	0.0493
bmi	0.04606	1.04714	0.05135	0.897	0.3697	0.1848
grade	-0.52443	0.59189	0.25849	-2.029	0.0425	0.4787
AAscore	2.83177	16.97543	1.89676	1.493	0.1355	0.0677
raceEA	-0.15651	0.85512	1.62874	-0.096	0.9234	0.0383
AAscore:race	0.94517	2.57324	2.38465	0.396	0.6918	0.3459

Revision Table 3: Different variations of Cox proportional models attempted. The row AAscore denotes the BA-community score.

6. Figure 3 also lacks clarity. What does the Delta Prob represent? For example, in panel 3a) the value is 0.26, yet there is an indication (***) that there is a significant difference, so this must not be a P-value? This is not standard notation. The survival curves indicate increased risk for the BA-specific combined interaction scores for both races, not just BA patients. Is this race specific?

Response: In response to this review concern, we have now included log-rank test-derived p values after adjusting for age, race and stage.

7. A spatial transcriptomic dataset was used to validate the proteins associated with the top 5

BA and WA cell-cell interactions identified. Protein names were converted to gene names, which were then queried in the spatial transcriptomic dataset. The transition from the original 20 IMC clusters to racially-specific 9 BA and 6 WA spatial cell-cell interactions, then to summed cell-cell interaction scores, then to proteins associated with the top 5 BA and WA cell-cell interactions, and finally to the 5 top gene queries, is difficult to follow. The manuscript would greatly benefit from a flowchart that summarizes these experiments and what samples/data are being used for each step. The manuscript then transitions to niche-specific gene signatures. Validation of the niches was done using both samples from an independent cohort, as well as from the original samples from the IMC analysis. It is unclear why the original samples are being used in the validation. Supplemental Tables 2 and 3 appear to be the same list of IMC antibodies, with the exception of 1-2 antibodies. Why are these both needed?

Response: We thank the reviewer for this question. We have now included new Figure 1b that describes a flow chart summarizing the experiments and describing what samples/data are used for each step. The primary validation of the niche-specific gene signatures were done using 10X ST data collected on independent set of 9 TNBC tumors from Baylor Scott and White (BSW2 validation) and 10 TNBC tumors from Georgia cohort (please refer to Figure 7). In addition, we have also performed nanostring GeoMx digital spatial profiling of the immune and epithelial compartment of the 57 TNBC tumors from the Baylor Scott and White Discovery data (which was examined using IMC in the discovery setting) to show that at the transcript level we still observe the enrichment of the BA community genes in the immune compartment. Supplementary Table 2-4 contain the list of antibodies and the metal tags used for the three independent IMC experiments involving Baylor Scott and White Discovery data, Baylor Scott and White Validation data and Roswell Park Validation data. As we moved along these validation experiments, we added additional antibodies into the panel to address reviewer concerns. Hence, we have used three supplementary tables to describe this information.

8. The manuscript would greatly benefit from improved organization and more clarity. In addition, the manuscript is overly dense with methods, data, results, figures, and tables. The progression from experiment to experiment is not completely clear, and often requires excessive searching between the methods, results and figures/tables to understand what was being done. Finally, given the many limitations, the conclusions drawn from this research could simply be an overinterpretation of the data. A more rigorous approach to validation of these findings is warranted.

Response: We have improved the organization and enhanced the clarity of the manuscript. We have also validated our findings using IMC analysis on an independent set of patient-derived TNBC samples from Roswell Park Comprehensive Cancer Center and included the results in the revised manuscript.

Point By Point Response to Review Comments

Reviewer #1 (Remarks to the Author):

General Comments:

The revised manuscript by Zhu et al. demonstrates substantial improvement through the inclusion of new analytical approaches and additional data. The authors have thoroughly addressed prior reviewer feedback, resulting in a more rigorous and impactful study. Most notably, the integration of imaging mass cytometry (IMC)–based validation of 46 triple-negative breast cancer (TNBC) tumors from Roswell Park Comprehensive Cancer Center enhances the robustness and reproducibility of their findings.

A major strength of the revised manuscript is the shift from analyzing individual cell-cell interactions (CCIs) to identifying higher-order spatial “communities” composed of multiple, functionally related CCIs. These communities were validated across three independent clinical cohorts, demonstrating greater resilience to institutional and biological variability compared to the previously used BA Q1–Q5 and WA Q1–Q5 CCI categorizations.

The manuscript’s emphasis on racially distinct tumor microenvironments (TMEs) in TNBC is both timely and highly relevant. The updated analyses provide stronger support for the claim that spatial organization within the TME contributes to prognostic differences.

Response: We thank the reviewer for the positive comments.

Nevertheless, several suggestions remain for further refinement:
Remaining Concerns and Recommendations

1. Improve Accessibility of Complex Content
The manuscript contains dense and technically complex spatial biology concepts, which may overwhelm readers despite its scientific depth.

Suggestions:

o Include a graphical abstract or schematic summarizing the analysis pipeline (e.g., IMC → Communities → ESGs → Prognosis → Validation).

Response: A graphical schematic summarizing IMC analysis pipeline is in Figure 1b. We have indicated Survival Association in Figure 1b that covers prognosis. We have also included Nanostring that covers ESG validation.

o Add a visual summary table outlining each cohort (e.g., BSW, Roswell, Georgia), the analysis performed (IMC, ST, GeoMX), key findings, and validation outcomes.

Response: A schematic outlining each cohort and analysis performed is in Figure 1 a. The key findings are summarized in Figure 8.

o Reorganize the Discussion section to highlight biological insights—such as BA-niche and WA-niche characteristics and their therapeutic implications—rather than reiterating results. Additionally, consider briefly addressing the clinical utility of niche-specific signatures for patient stratification or trial design.

Response: The above is now included in the revised discussion.

o The discussion should also contextualize discrepancies with previous findings regarding the prevalence of exhausted T cells in Black patients (e.g., Yao et al., JNCI 2021, also from Roswell Park investigators). Clarify whether these differences are attributable to the more granular niche-based analysis used here.

Response: The above reference is now cited as reference 15 in the revised discussion.

2. Clarify the Added Prognostic Value of Spatial Context Over Cell Abundance While the rebuttal and Supplementary Table 1 highlight that single-cell abundance alone lacks prognostic value, this critical point could be more explicitly presented. Suggestions:

o Introduce a dedicated subsection in the Results titled “Spatial Interactions Add Prognostic Value Beyond Cell Abundance.”

o Provide a concise explanation of why spatial proximity—rather than mere immune cell presence—better predicts outcomes, particularly in BA niches.

Response: We have included the above in revised Discussion. Specifically, to address the second concern, we have included the following paragraph in the revised discussion:

“Crucially, our analysis indicates that these spatial interactions and community structures provide prognostic information beyond what simple cell abundance reveals. As demonstrated (Supplementary Table 3), single-cell abundance alone often lacked significant prognostic value, whereas the spatial context—reflecting the proximity

required for direct signaling and localized immune modulation, particularly within the suppressive BA niches—proved significantly associated with survival outcomes.”

3. Refine Interpretation of Immune Phenotypes (Suppl. Fig. 19)

The comparative analysis of BA/WA niche-enriched signature genes (ESGs) with inflamed/excluded/ignored immune phenotypes is insightful but remains correlative. Suggestion:

o In the Discussion, clarify that the observed alignment with known immune phenotypes is hypothesis-generating and warrants functional validation (e.g., assessment of immune signaling activity or cytokine profiling).

Response: This is now included in the revised Discussion. Specifically, to address this concern, we have included the following paragraph in the revised discussion:

“Future functional validation, potentially through assessment of immune signaling activity or localized cytokine profiling within these niches, is warranted to confirm these phenotypic characterizations.”

4. Address Potential Socioeconomic Confounders

The prognostic differences observed in WA-Comm1 between the Baylor and Roswell cohorts may be influenced by underlying socioeconomic factors, including food insecurity or access to care.

Suggestion:

o If socioeconomic data are not available, include a statement acknowledging this limitation and discussing how such factors may contribute to cohort differences in immune function and outcomes.

Response: This is now included in the revised Discussion. Specifically, to address this concern, we have included the following paragraph in the revised discussion:

“Addressing these socioeconomic factors would be an area of future investigation where such data are available.”

Minor Comment

- The conclusion refers to unique immune “niches” in tumors from white patients. However, the data appear to reflect generalized immune infiltration rather than distinct spatial congregation into niches. Clarification of this distinction would improve the precision of the conclusions.

Response: We used the term unique immune niches in tumors from white patients to indicate that this is not a random immune infiltration event. It is specific to the immunotype cell types mentioned in the manuscript. Also, the term 'niche' in this context does not imply a uniform size; for example, the BA niche demonstrates a more extensive spatial clustered organization, while the WA niche is confined to smaller neighborhoods of interacting cells that are less organized.

Reviewer #2 (Remarks to the Author):

Response: Thank you

Reviewer #3 (Remarks to the Author):

I have reviewed the revision of the manuscript “Integrative spatial omics reveals distinct tumor-promoting multicellular niches and immunosuppressive mechanisms in Black American and White American patients with TNBC”. In the first review round, I pointed out that the manuscript was lengthy and suggested to trim it down to the essentials. The manuscript is not shortened significantly, however, it is now written in a more precise manner. Especially the description of the cohorts is improved, and illustrations of the cohort and available data are included so it is easier to understand what has been done. I still think that the number of figures is excessive. Most of them are illustrative, however figure 3 and suppl. Figure 9 are very difficult to interpret and does not illustrate well the cell communities. I would consider revising these.

Response: We thank the reviewer for the positive comments. We have included additional details in the legends for Figure 3 and Supplementary Figure 9 to help understand the cell communities better.

It is not possible to appreciate the ROI segmentation into AOIs for the geomx data in suppl. figure 2.

Response: We have included high-resolution images showing segmentation of ROI into AOIs for the geomx in the supporting data folder.

Reviewer #3 (Remarks on code availability):

The code is available on GitHub. I lack competency in python, therefore, I can not review the quality of the code and the reproducibility.

Response: All the codes are in GitHub.

Reviewer #4 (Remarks to the Author):

The authors have made extensive revisions to the manuscript, which have enhanced its clarity and quality. They have also satisfactorily addressed the concerns that I previously raised.

Response: We thank the reviewer for the positive comments.